# The integrated stress response remodels the microtubule-organizing center to clear unfolded proteins following proteotoxic stress

Brian Hurwitz[1†], Nicola Guzzi[1†], Anita Gola[1], Vincent F Fiore[1], Ataman Sendoel[2], Maria Nikolova[1], Douglas Barrows[3], Thomas S Carroll[3], H Amalia Pasolli[4], Elaine Fuchs[1]*

[1]Howard Hughes Medical Institute, Robin Chemers Neustein Laboratory of Mammalian Cell Biology and Development, The Rockefeller University, New York, United States; [2]Institute for Regenerative Medicine, University of Zurich, Zurich, Switzerland; [3]Bioinformatics Resource Center, The Rockefeller University, New York, United States; [4]Electron Microscopy Resource Center, The Rockefeller University, New York, United States

*For correspondence:
fuchslb@rockefeller.edu

[†]These authors contributed
equally to this work

**Abstract** Cells encountering stressful situations activate the integrated stress response (ISR) pathway to limit protein synthesis and redirect translation to better cope. The ISR has also been implicated in cancers, but redundancies in the stress-sensing kinases that trigger the ISR have posed hurdles to dissecting physiological relevance. To overcome this challenge, we targeted the regulatory node of these kinases, namely, the S51 phosphorylation site of eukaryotic translation initiation factor eIF2α and genetically replaced eIF2α with eIF2α-S51A in mouse squamous cell carcinoma (SCC) stem cells of skin. While inconsequential under normal growth conditions, the vulnerability of this ISR-null state was unveiled when SCC stem cells experienced proteotoxic stress. Seeking mechanistic insights into the protective roles of the ISR, we combined ribosome profiling and functional approaches to identify and probe the functional importance of translational differences between ISR-competent and ISR-null SCC stem cells when exposed to proteotoxic stress. In doing so, we learned that the ISR redirects translation to centrosomal proteins that orchestrate the microtubule dynamics needed to efficiently concentrate unfolded proteins at the microtubule-organizing center so that they can be cleared by the perinuclear degradation machinery. Thus, rather than merely maintaining survival during proteotoxic stress, the ISR also functions in promoting cellular recovery once the stress has subsided. Remarkably, this molecular program is unique to transformed skin stem cells, hence exposing a vulnerability in cancer that could be exploited therapeutically.

## Editor's evaluation

This interesting study identifies why or how the integrated stress response pathway regulates cell recovery upon proteotoxic stress, which is especially interesting in cancer cells resistant to proteasome inhibitors. The authors conclude that translation initiation of mRNAs encoding microtubule cytoskeleton, centrosome, and ATF5 proteins is necessary to recover from proteotoxic stress. This article makes a strong contribution to the literature.

## Introduction

Eukaryotic cells rely on highly conserved pathways to dynamically control translational machinery in response to various stimuli. One such pathway, the integrated stress response (ISR), is triggered when one of four stress-sensing kinases becomes active: (1) heme-regulated inhibitor kinase (HRI) is activated by oxidative stress, such as arsenite; (2) protein kinase R (PKR) is activated by viral infections; (3) PKR-like endoplasmic reticulum kinase (PERK) is induced when unfolded proteins accumulate (proteotoxic stress) and when the endoplasmic reticulum (ER) becomes stressed; and (4) general control non-depressible 2 (GCN2) is induced in poor nutrient conditions, particularly amino acid deprivation (*Costa-Mattioli and Walter, 2020*; *El-Naggar and Sorensen, 2018*). Activation of any of these four kinases results in phosphorylation of serine 51 of eIF2α, a core component of the canonical translational initiation complex. Since phosphorated eIF2α blocks the guanine nucleotide exchange factor eIF2B from stimulating the eIF2-GTP-methionyl-initiator tRNA ternary complex, the translation of housekeeping mRNAs is decelerated, and initiation of new protein synthesis is dampened. Concomitantly, non-canonical translational mechanisms emerge to synthesize key stress response proteins that help restore physiological balance to the cell and aid in survival (*Figure 1A*; *Pakos-Zebrucka et al., 2016*).

This paradoxical translational upregulation of a subset of mRNAs can be driven by several mechanisms, including (1) lowering the dependency on the cap-binding complex eIF4F (*Schuster and Hsieh, 2019*), (2) initiating non-canonical translation within 5′ untranslated regions (5′ UTRs) that harbor dynamic N6-methyladenosine (m6A) RNA modification sites and/or 5′ upstream open reading frames (uORFs) (*Sendoel et al., 2017*; *Starck et al., 2016*; *Zhou et al., 2018*), and finally (3) directly recruiting translation factors and ribosomes to the 5′ UTR of stress-responsive mRNAs (*Guan et al., 2017*). This selective repurposing of the translational machinery forms the foundation of a cellular stress response aimed at restoring homeostasis.

The best-studied translational target of the ISR is *Atf4*, encoding activating transcription factor 4 (ATF4), whose translation is repressed in homeostatic conditions by the presence of inhibitory uORFs in its 5′ UTR. Upon stress, *Atf4* translation is unleashed, and the newly synthesized protein mediates transcriptional upregulation of stress response genes, thus acting as a crucial regulator in the balance of cell death and survival. In fact, known ATF4 targets include both cytoprotective and apoptotic factors. That a single ISR target can have such diverging and context-specific roles highlights the complexity of the cellular stress response and indicates the possibility that the ISR might translationally regulate additional cellular programs to coordinate survival, adaptation, and recovery from stress.

At the interfaces of cellular proliferation, apoptosis, survival, and protein synthesis, the ISR has naturally emerged as an area of interest in cancer research. Although studies on the role of the ISR in cancer have been conflicting, the notion that the ISR is protective for cancer cells has gained traction in recent years (*Ghaddar et al., 2021*; *Koromilas, 2015*). This is in line with the observation that cancer cells experience hostile microenvironments characterized by nutrient deprivation and low oxygen availability. Additionally, the proliferative stress and elevated metabolic demands of cancer cells create an increased reliance on the cellular mechanisms that maintain proteostasis, a central function of the ISR (*Cubillos-Ruiz et al., 2017*). Indeed, proteasome inhibition as an anticancer therapy is aimed at exploiting this vulnerability. That said, although proteasome inhibitors have become the standard of care for multiple myeloma and mantle-cell lymphoma (*Manasanch and Orlowski, 2017*), this strategy has been less effective in solid tumors, suggesting that these tumors may utilize mechanisms that enable them to cope, survive, and recover in the face of proteotoxic stress, thereby evading therapeutics (*Tian et al., 2021*).

In this study, we focus on the most common and life-threatening solid tumors, squamous cell carcinomas (SCCs), which affect stratified squamous epithelia of the skin, esophagus, lung, and head and neck. To investigate the roles of the ISR in this cancer, we took advantage of our ability to culture the tumor-initiating stem cells from mouse skin SCCs and generated primary, clonal cell lines in which the endogenous eIF2α alleles were replaced by myc-epitope tagged but otherwise fully functional versions of either wild-type eIF2α or eIF2α-S51A (ISR-null). Although unable to mount an ISR in the face of stress, ISR-null SCC stem cells formed tumors that were similar in morphology and proliferation characteristics to controls. However, both in vivo and in vitro, when challenged with proteotoxic stress, ISR-null SCC cells fared considerably more poorly than their ISR-competent counterparts.

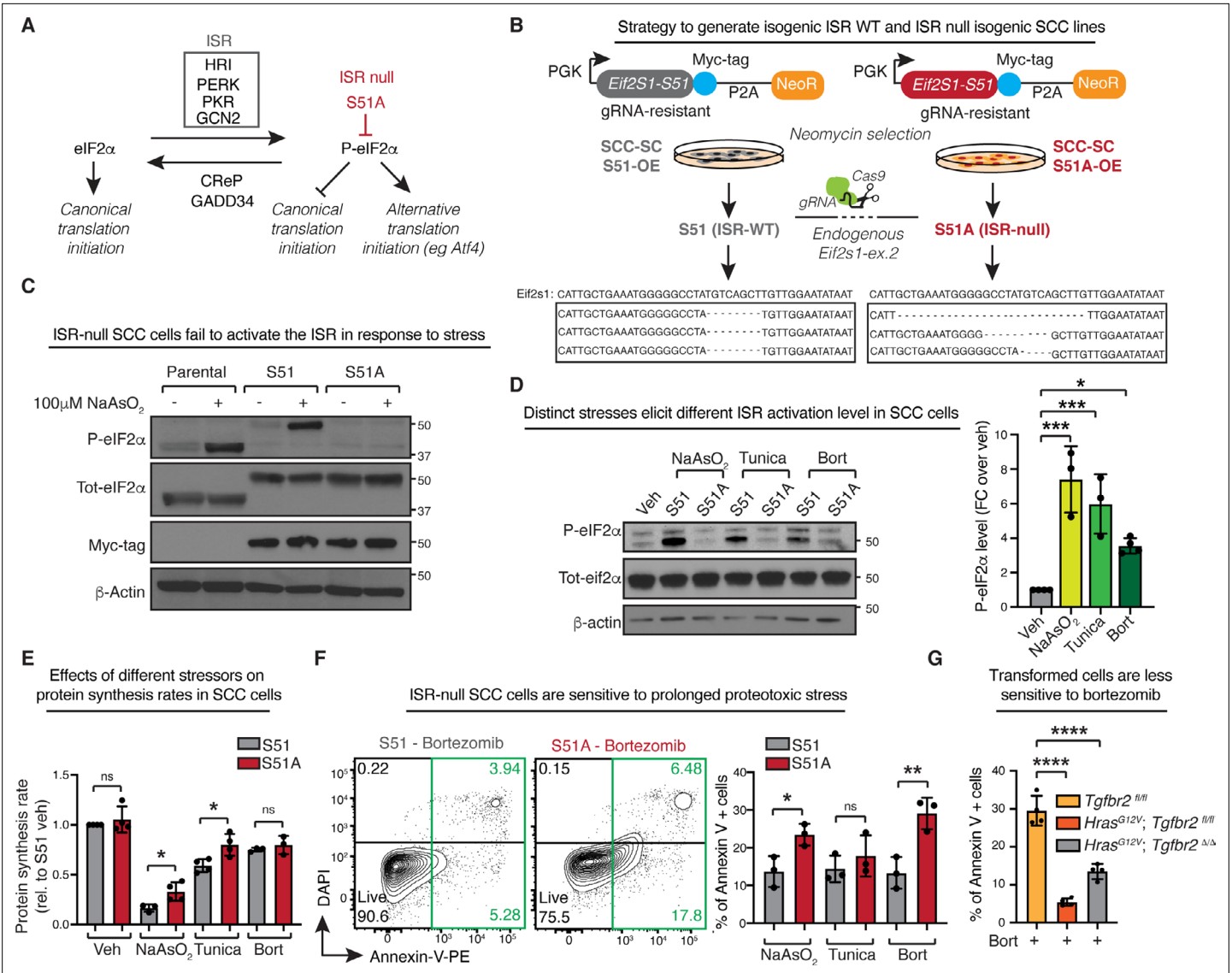

**Figure 1.** The integrated stress response (ISR) induces a pro-survival program in squamous cell carcinoma (SCC) cells upon exposure to proteotoxic stress. (**A**) Schematic of the integrated stress response pathway showing activating kinases, inhibitory phosphatases, and downstream effects. Mutation of eIF2α serine 51 to alanine renders cells insensitive to the ISR activating kinases ('ISR-null'). (**B**) Reconstitution and knockout strategy to generate ISR-null SCC cell lines. Constructs encoding a myc-epitope tagged and CRISPR-Cas9-resistant eIF2α (either S51 or S51A) are packaged into lentivirus and integrated into a parental SCC stem cell (SCC-SC) line. Following neomycin selection, CRISPR-Cas9 is delivered by RNP transfection to selectively knock out endogenous eIF2α alleles. Clones grown from single cells are analyzed for knockout by deep sequencing the gRNA target locus. (**C**) Representative immunoblot confirms both the replacement of the endogenous eIF2α with myc-tagged eIF2α and the failure of cells expressing eIF2α-S51A to activate the ISR. Cells were treated with 100 μM sodium arsenite for 6 hr to induce oxidative stress. (**D**) ISR-competent cells induce eIF2α phosphorylation upon stress. Representative immunoblot shows levels of eif2α phosphorylation of cells treated with either 50 μM sodium arsenite, 100 ng/mL tunicamycin, or 100 nM bortezomib for 6 hr. Bar graph shows normalized levels of eIF2a phosphorylation ± SD in three independent biological experiments. *p<0.05, ***p<0.001 (*t*-test). (**E**) Bar graph shows normalized protein synthesis rates ± SD relative to vehicle-treated S51 SCC cells in S51 and S51A cells treated with either 50 μM sodium arsenite, 100 ng/mL tunicamycin, or 100 nM bortezomib for 6 hr in at least three independent biological experiments. *p<0.05, ns, no statistical significance (*t*-test). (**F**) ISR-null SCC cells are more sensitive to stress. Early signs of apoptosis are measured by Annexin-V binding in control and ISR-null cells treated with 5 μM sodium arsenite, 100 ng/mL tunicamycin, or 4 nM bortezomib for 24 hr. Bar graph shows percentage of Annexin-V-positive cells ± SD of N = 3 independent biological experiments. *p<0.05, **p<0.01, ns, no statistical significance (*t*-test). (**G**) Keratinocytes isolated from littermate controls were treated with 4 nM bortezomib for 24 hr. Wild-type (*Tgfbr2*<sup>fl/fl</sup>), pre-oncogenic (*Hras*<sup>G12V</sup>; *Tgfbr2*<sup>fl/fl</sup>), and SCC keratinocytes (*Hras*<sup>G12V</sup>; *Tgfbr2*<sup>Δ/Δ</sup>) were compared. Bar graph shows percentage of Annexin-V-positive cells ± SD in four independent biological replicates. ***p<0.001 (*t*-test).

The online version of this article includes the following source data and figure supplement(s) for figure 1:

*Figure 1 continued on next page*

*Figure 1 continued*

**Source data 1.** Original immunoblots show the characterization of S51 and S51A squamous cell carcinoma (SCC) cell lines (related to *Figure 1C*).

**Source data 2.** Original immunoblots show phosphorylation status of eIF2α upon different stresses (related to *Figure 1D*).

**Figure supplement 1.** Effect on different stresses on protein synthesis rates and mTORC1 activation in integrated stress response (ISR)-competent and ISR-null cells.

**Figure supplement 1—source data 1.** Original immunoblots show protein synthesis rates upon different stresses (related to *Figure 1—figure supplement 1A*).

**Figure supplement 1—source data 2.** Original immunoblots show components of the mTORC1 pathway upon different stresses (related to *Figure 1—figure supplement 1B*).

**Figure supplement 2.** Integrated stress response (ISR)-null cells are more sensitive to stress but show no difference at steady state.

**Figure supplement 3.** Transformed keratinocytes are less sensitive to proteotoxic stress.

While the HRI and PERK kinases have emerged as major players in sensing misfolded proteins (*Abdel-Nour et al., 2019*; *Harding et al., 1999*), a detailed understanding of how the ISR promotes proteostasis is lacking. To probe into the mechanisms that link the ISR to proteostasis, we coupled genetics, cell biology, pharmacological inhibitors, ribosomal profiling, and finally functional analyses. We traced the connection to a group of centrosomal proteins that become selectively translated in response to proteotoxic stress. We show that these proteins act by strengthening the organizing center for microtubule dynamics that are needed to efficiently amass unfolded proteins at the peri-centrosomal locale, where they can be efficiently targeted for destruction and clearance. At the crux of this response is ATF5, a core component of the centrosome, which we found to be translationally upregulated selectively upon proteotoxic stress. Here, we unravel a cancer-specific mechanism in which ATF5 protects the microtubule-organizing center (MTOC) to promote cellular recovery and resistance to proteotoxic stress. Our findings add a new dimension, microtubule dynamics, to the role of the ISR not only in stress, but also in the recovery of cells to stress. In so doing, our findings also expose a hitherto unappreciated vulnerability of cancer cells that is unleashed when they are unable to mount an ISR in the face of proteotoxic stress.

## Results

### Activation of the ISR upon proteotoxic stress promotes survival in SCC cells

In order to directly test the role of the ISR in SCC cells, we generated eIF2α-S51A or 'ISR-null' cancer cells using a knockout and reconstitution strategy. For this purpose, we used an aggressive *Hras^G12V* murine, primary-derived skin SCC line expressing an eGFP reporter (*Yang et al., 2015*). We transduced these cells with lentiviral constructs harboring a PGK promoter-driven cDNA encoding either the wild-type (S51) or phospho-dead (S51A) eIF2α protein (encoded by the *Eif2s1* locus) (*Figure 1B*). The *Eif2s1* transgenes each contained a synonymous mutation in a protospacer adjacent motif (PAM) site that rendered it resistant to a guide RNA (gRNA) that could be used to specifically target the endogenous *Eif2s1* gene for CRISPR/Cas9 deletion.

Cells transduced with the myc-tagged *Eif2s1* constructs were transfected with liposomes harboring CRIPSR-Cas9/sgRNA ribonucleoproteins, which targeted the ablation of the endogenous *Eif2s1* alleles, and thus left the myc-tagged S51 or S51A transgenes as the sole source of eIF2α expression. Following ablation and reconstitution, single cells were isolated by fluorescence-activated cell sorting (FACS) and used to generate stable SCC clones. Successful targeting of the two endogenous *Eif2s1* alleles was verified by genomic sequencing (*Figure 1B*).

Two of each S51 and S51A eIF2α replacement clones were chosen for further study. In all assays presented in *Figures 1 and 2*, the clones of the same eIF2α status grew and behaved similarly. Hence for the purposes of this study, we show the results on pooled clones displaying common genotypes. Immunoblot analyses revealed that total eIF2α levels were comparable to the parental eIF2α clone, and the replacement eIF2α proteins exhibited the expected increase in size due to the epitope tag (*Figure 1C*). Importantly, when we treated these SCC lines with sodium arsenite to induce oxidative stress and activate the HRI kinase (*Sendoel et al., 2017*), we observed that like the parental clone, the

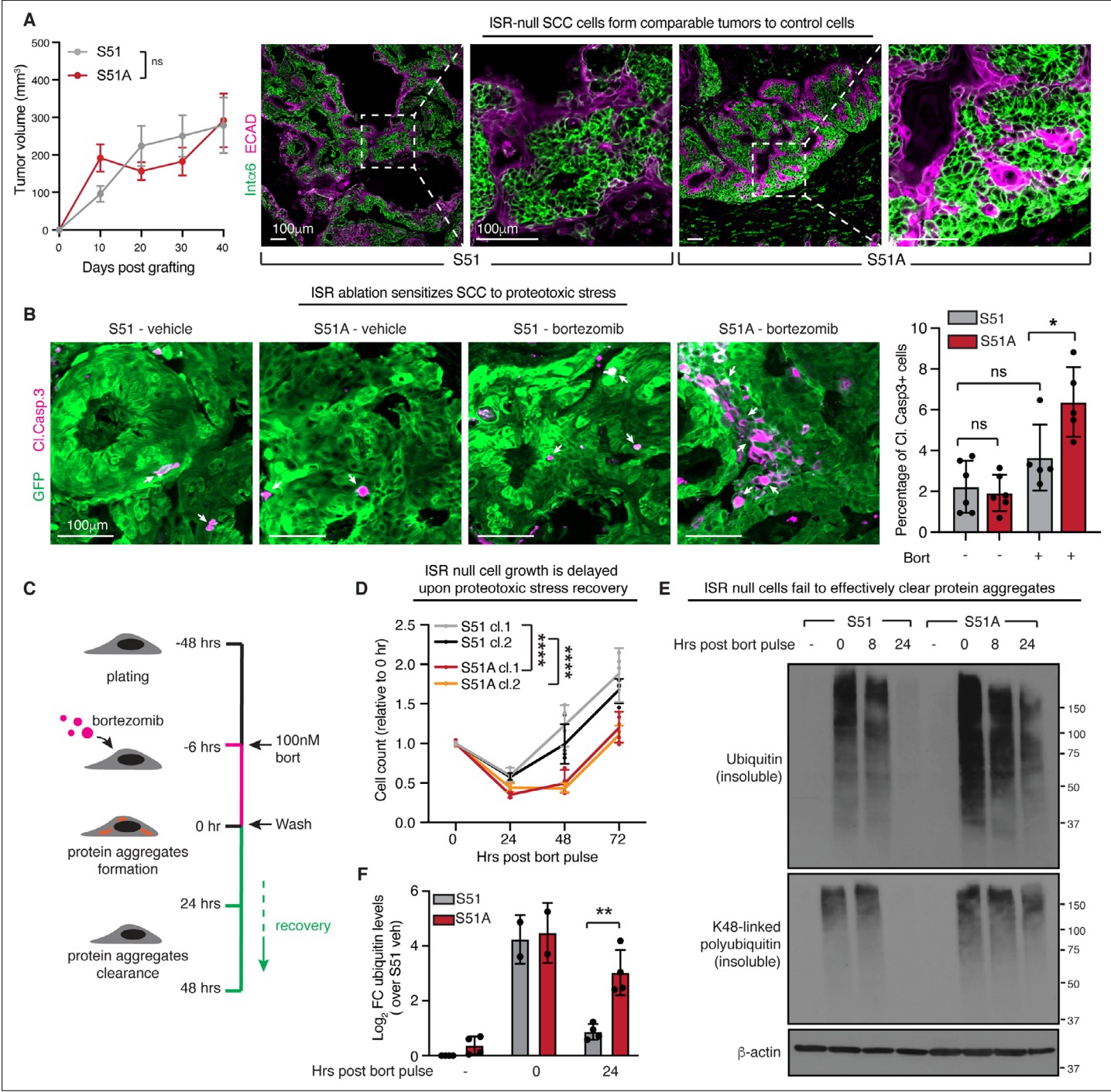

**Figure 2.** The integrated stress response (ISR) is required for efficient clearance of protein aggregates. (**A**) Left: growth of tumor allografts derived from control or ISR-null squamous cell carcinoma (SCC) cells. Tumor volume is comparable between conditions. Error bars represent SEM from N = 16 tumors per condition. ns, no statistical significance (simple linear regression). Right: α6-integrin and E-cadherin immunofluorescence of SCC on day 17 reveals that ISR-ablation does not impact tumor morphology. Boxed regions are shown at higher magnification images to the right of each image. Scale bar 100 μm. (**B**) Cleaved caspase 3 immunofluorescence of tumor sections reveals elevated apoptosis, a sign of increased sensitivity of ISR-null SCC tumors to proteotoxic stress (bortezomib treatment 1.2 mg/kg i.p. on days 13, 15, and 17). No significant difference in viability is observed in vehicle-treated SCC tumors. White arrows indicate cleaved caspase 3-positive cells. Bar graph on the right shows percentage of cleaved caspase 3-positive cells ± SD within the GFP+ tumor cells, in at least five independent tumors. * p<0.05, ns, no statistical significance (one-way ANOVA, multiple comparisons). Scale bar 100 μm. (**C**) Schematic of protein aggregate pulse-recovery experiments. Cells are treated with a saturating (100 nM) dose of bortezomib for 6 hr, a timepoint at which protein aggregates have accumulated, but cells are still viable. Bortezomib is washed from the media and cells are then monitored during the recovery phase. (**D**) Cellular proliferation during the protein aggregate recovery phase indicates that control cells regain proliferative capacity

*Figure 2 continued on next page*

*Figure 2 continued*

24 hr prior to ISR-null cells. N = 4 biological replicates, n = 16 total technical replicates, error bars denote SD of technical replicates. ****p<0.0001 (two-way ANOVA, multiple comparisons). (**E**) Protein aggregates evaluated by anti-ubiquitin and K48-linked polyubiquitin immunoblot following cellular fractionation of soluble and insoluble proteins. Representative immunoblot of the insoluble fraction shows clearance of aggregates in control but not ISR-null cells at 24 hr. (**F**) Bar graph shows log$_2$ fold change (FC) ubiquitin levels ± SD normalized over β-actin and relative to vehicle-treated control cells in at least two independent biological experiments. **p<0.01 (*t*-test).

The online version of this article includes the following source data and figure supplement(s) for figure 2:

**Figure supplement 1.** Characterization of integrated stress response (ISR)-null squamous cell carcinoma (SCC) tumors.

**Figure supplement 1—source data 1.** Original immunoblots show phosphorylation status of eIF2α in cells isolated from allografted tumors (related to *Figure 2—figure supplement 1B*).

**Figure supplement 2.** Characterization of bortezomib pulse and recovery system.

**Figure supplement 3.** Characterization of protein aggregate source and clearance mechanism.

**Figure supplement 3—source data 1.** Original immunoblots show ubiquitin levels in the insoluble fraction of S51 cells treated with cycloheximide or harringtonine and following recovery from proteotoxic stress (related to *Figure 2—figure supplement 3A*).

**Figure supplement 3—source data 2.** Original immunoblots show ubiquitin levels in the insoluble fraction of S51 cells treated with salubrinal and following recovery from proteotoxic stress (related to *Figure 2—figure supplement 3B*).

**Figure supplement 3—source data 3.** Original immunoblots show ubiquitin levels in the insoluble fraction of S51 cells treated with bafilomycin A1 and following recovery from proteotoxic stress (related to *Figure 2—figure supplement 3C*).

**Figure supplement 3—source data 4.** Original immunoblots show ubiquitin levels in the insoluble fraction of S51 cells treated with PERK inhibitor and following recovery from proteotoxic stress (related to *Figure 2—figure supplement 3D*).

---

stressed eIF2α S51 cells displayed phospho-S51 immunolabeling, while the eIF2α S51A clones were refractory to phosphorylation.

Taken together, these results underscored the efficacy of our knockout and replacement strategy, and verified the complete loss of eIF2α phosphorylation at the heart of the ISR.

To further interrogate the functionality of our SCC lines in mounting an ISR, we analyzed eIF2 responsiveness to different types of stress. In all cases, our S51 cells showed a marked increase in phosphorylation following stress induction, albeit at different levels in response to distinct stresses (*Figure 1D*). As eIF2α phosphorylation decommissions this canonical translational regulator and dampens protein synthesis rates, we also examined translational rates in response to the three stressors. Using puromycin incorporation as a gauge to measure nascent protein synthesis, we observed that indeed all three stressors dampened protein synthesis rates in our eIF2α S51 SCC cells (*Figure 1E*, *Figure 1—figure supplement 1A*). As expected, no ISR-dependent differences in protein synthesis were observed in unstressed cells. However, stressed S51A SCC cells failed to limit protein synthesis at the same rate as S51 SCC cells following exposure to oxidative stress or tunicamycin (which induces ER-stress through eIF2α-S51 phosphorylation by PERK kinase) (*Figure 1E*, *Figure 1—figure supplement 1A*).

By contrast, the proteosome inhibitor bortezomib (a boronic acid dipeptide derivative) seemed refractory to ISR-dependent changes in protein synthesis rates. This was surprising as bortezomib acts by accumulating misfolded proteins, which in turn induce ER-stress and eIF2α-S51 phosphorylation (*Figure 1D*; *Jiang and Wek, 2005*; *Suraweera et al., 2012*; *Yerlikaya and Okur, 2020*). Suspecting alternative mechanisms, we first showed that bortezomib inhibited the mTORC1 pathway (*Figure 1—figure supplement 1B*), suggesting that mTORC1 is primarily responsible for the global reduction in protein synthesis rates during proteotoxic stress response.

We next asked whether the ISR might be dispensable in our SCC cells exposed to proteotoxic stress. Under the serum-rich culture conditions used here, S51 and S51A SCC cells proliferated at comparable rates and showed no differences in viability (*Figure 1—figure supplement 2A and B*). Analogously, the morphologies of these cells were indistinguishable under these conditions (*Figure 1—figure supplement 2C*). However, upon exposure to proteotoxic stress, nearly 3× more eIF2α-S51A cells were positive for the apoptotic marker Annexin-V over counterparts with an intact ISR. This sensitivity was even greater for bortezomib than for other stressors (*Figure 1F*, *Figure 1—figure supplement 2D*). These findings unequivocally established the importance of the ISR in SCC cells faced with proteotoxic stress. However, its role in the process was not in dampening global protein synthesis, but rather in inducing a pro-survival program.

Was this feature unique to transformed cells? To this end, we compared sensitivity to bortezomib in skin keratinocytes isolated from littermates: either wild-type control (*Tgfbr2^fl/fl^*), pre-oncogenic (*Hras^G12V^; Tgfbr2^fl/fl^*), or fully transformed (*Hras^G12V^; Tgfbr2^Δ/Δ^*). Interestingly in the presence of bortezomib, pre-oncogenic and fully transformed stem cells were significantly more resistant to cell death compared to wild-type stem cells (*Figure 1G*, *Figure 1—figure supplement 3A*). Taken together, our collective data indicate that when faced with proteotoxic stress, SCC cells gain a fitness advantage as long as they can activate an ISR. Our data further implied that in proteotoxic stress the ISR's role in cancer cells seems to be to shift to a translational landscape that is more favorable for survival.

## The ISR promotes efficient clearance of protein aggregates

When epidermal stem cells acquire a single oncogenic mutation and will eventually progress to SCC, they activate alternative translational pathways suggestive of an ISR (*Blanco et al., 2016*; *Sendoel et al., 2017*). Intriguingly, however, when we injected our SCC cells into the skins of immunocompromised, athymic (*Nude*) mice, S51A and S51 SCC cells both formed tumors that grew similarly and showed similar morphologies (*Figure 2A*, *Figure 2—figure supplement 1A*). This was not attributable to 'escapers' that had somehow circumvented the S51A mutation in vivo as FACS-sorted SCC stem cells from S51A tumors still displayed insensitivity to sodium arsenite-induced eIF2α phosphorylation in contrast to their S51 counterparts (*Figure 2—figure supplement 1B*).

Strikingly, and consistent with our in vitro findings, when we treated our mice with bortezomib, the in vivo S51A SCC tumors exhibited significantly greater sensitivity to proteotoxic stress and apoptosis than their counterparts (*Figure 2B*). These findings were particularly relevant given that despite intense interest in proteasome inhibitors as a potential new line of cancer therapeutics, solid tumors have shown resistance to these drugs (*Fournier et al., 2010*; *Manasanch and Orlowski, 2017*). Taken together, our results raised the tantalizing possibility that, if the ISR is crippled in solid tumors, their tumor-propagating stem cells may acquire increasing sensitivity to added stress.

While the ISR was known to be activated by unfolded proteins that accumulate upon proteasome inhibition (*Figure 1D*; *Pakos-Zebrucka et al., 2016*), it remained unclear how the ISR was protecting cancer cells against proteotoxic stress. To dissect the underlying mechanisms, we asked whether the ISR might participate in the cell recovery phase. To test this possibility, we established a two-step in vitro model of first triggering SCC cells to accumulate an excess of unfolded proteins and then allowing them to recover from the stress. To this end, we treated cultured control or ISR-null SCC cells in vitro with a saturating dose of bortezomib for 6 hr, at which point we washed the cells and switched to fresh media to allow cells to recover (*Figure 2C*). Following this bortezomib 'pulse,' control SCC cells with a competent ISR recovered and began to proliferate within 24–48 hr. In striking contrast, ISR-null SCC cells took a full 24 hr longer than their counterparts before they began to proliferate again (*Figure 2D*). Intriguingly, this could not be imputed to differences in viability on a short timescale since following this treatment regime, ISR-null and ISR-competent SCC stem cells displayed no difference in Annexin-V positivity either at 6 hr of bortezomib treatment or at 24 hr after recovery (*Figure 2—figure supplement 2A–C*).

The first step in targeting unfolded proteins for degradation is their ubiquitination (*Smith et al., 2011*). We therefore examined the clearance of ubiquitinated proteins in SCC cells recovering from proteotoxic stress. Following treatment with bortezomib, cells were lysed in radio-immunoprecipitation assay (RIPA) buffer, and soluble and insoluble proteins were then fractionated by centrifugation. Each fraction was normalized based on the protein concentration of the soluble fraction and then subjected to polyacrylamide gel electrophoresis and analyzed by anti-ubiquitin immunoblots.

ISR-null and control SCC cells responded similarly to bortezomib, displaying a rapid jump in ubiquitinated proteins within 6 hr of treatment (*Figure 2E and F*). After withdrawing bortezomib, however, the rate at which ubiquitinated proteins were cleared from the insoluble fraction was remarkably reduced in the ISR-null cells compared to control SCC cell lysates. As lysine 48-linked polyubiquitin is the specific mark of proteins targeted for proteasomal degradation (*Thrower et al., 2000*), we performed anti-K48-polyubiquitin immunoblot analyses (*Figure 2E*). These data confirmed that SCC cells that are unable to mount an ISR are not defective in their E3-ubiquitin-ligase system per se, but rather are impaired in their ability to efficiently clear proteins that are marked for proteosomal clearance.

We also used this assay to interrogate the source of the insoluble protein aggregates induced by bortezomib. To do so, we treated cells concurrently with cycloheximide, to block new protein synthesis and with bortezomib, to inhibit proteosome-mediated aggregated protein clearance. Inhibiting translation elongation with cyclohexamide or translation initiation with harringtonine almost entirely blocked the buildup of protein aggregates following 6 hr of bortezomib treatment. These data demonstrate that the aggregates in SCC cells treated with bortezomib originate from the accumulation of newly synthesized, unfolded proteins (*Figure 2—figure supplement 3A*). Additionally, we found that the ability of a cell to resolve the ISR was required to promote clearance of protein aggregates. Thus, when ISR-competent cells were treated with salubrinal, a selective inhibitor of eIF2α dephosphorylation, aberrant ubiquitinated aggregates accumulated following 24 hr of recovery from the bortezomib pulse (*Figure 2—figure supplement 3B*).

We next addressed the extent to which the proteosome versus autophagy is responsible for the clearance of these ubiquitin-marked protein aggregates during the recovery phase following bortezomib treatment. To do so, we treated with bortezomib for 6 hr, and then after washing out the drug, we either allowed both pathways to participate in clearance or added bafilomycin A1 to block the autophagy pathway. As shown in *Figure 2—figure supplement 3C*, BafA1 delayed recovery partially but not fully. Taken together, these data indicated that both autophagy and the proteasome cooperate in clearing these ubiquitin-marked protein aggregates during the recovery phase following proteotoxic stress.

Intriguingly, PERK kinase was hyperactivated in our S51A cells (*Figure 2—figure supplement 3D*). Given that PERK has other targets besides eIF2α S51, we asked whether and how PERK signaling might be involved in clearing these protein aggregates. Indeed, inhibition of PERK signaling by GSK2606414 resulted in increased aggregate accumulation, but this appeared to be irrespective of the status of eIF2α. PERK participated in protein aggregate clearance independently of the ISR, suggesting that other PERK targets, possibly NRF2 (*Pajares et al., 2017*), may be involved. That said, it was notable that even when PERK was inhibited, ISR-null cells retained a higher accumulation of ubiquitin-positive aggregates when compared to ISR-competent cells (*Figure 2—figure supplement 3D*). These data suggest that the phenotype observed is driven by an ISR-dependent mechanism that does not require PERK.

## The ISR responds to proteotoxic stress by upregulating translation of centrosomal proteins

We next turned to explore how the ISR might be involved in the process of cellular recovery from proteotoxic stress. The delayed rate in protein aggregate clearance seen in ISR-null cells was not attributable to a higher rate of global protein synthesis as this was comparable to the ISR-competent control cells (*Figure 1E*, *Figure 1—figure supplement 1A*). We therefore asked whether the ISR might be required to drive translation of select mRNAs upon proteotoxic stress, and if so, whether the proteins produced under such circumstances might give us clues into how the ISR functions in recovery. To this end, we performed ribosome profiling to landscape the ISR-mediated impact on translation during proteotoxic stress (*Figure 3A*; *Ingolia et al., 2009*; *McGlincy and Ingolia, 2017*).

Briefly, we subjected control and ISR-null SCC cells to proteotoxic stress (bortezomib) or a vehicle control. Quadruplicate samples of each condition were then lysed in the presence of cycloheximide in order to preserve the ribosome location along transcripts. Total mRNAs were saved, and the remaining lysates were treated with RNase-I to digest away all mRNAs that were not protected by ribosomes. Total mRNAs and the ribosome-protected fragments (RPFs) were then prepared for deep sequencing, and the translational efficiency (TE) was analyzed by assessing the ratio of RPFs to total mRNA reads, genome-wide.

Reads were aligned to the mouse reference genome, and quality control was performed to confirm that we had successfully purified and sequenced RNA fragments that were protected by actively translating ribosomes. Technical replicates were found to co-vary within samples by subjecting them to principal component analysis (*Figure 3—figure supplement 1A*). As a second quality control, metagene analysis was performed, which demonstrated that the majority of RPFs were found within coding segments, with relative peaks at the translation start and stop sites, as would be expected from a high-quality ribosome profiling dataset (*Figure 3—figure supplement 1B*). The read lengths peaked between 29 and 32 nucleotides, corresponding to the correct length

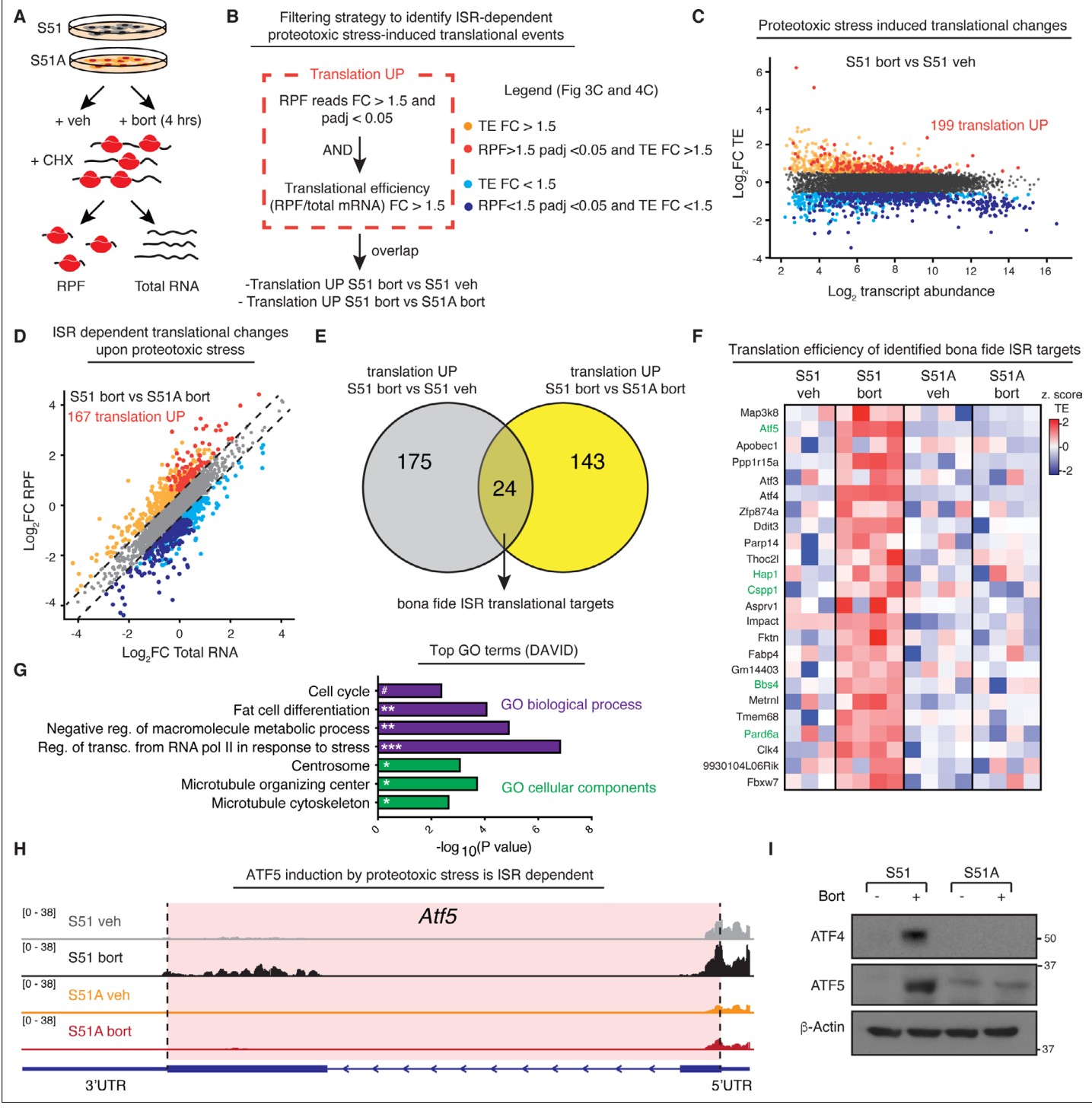

**Figure 3.** The integrated stress response (ISR) induces synthesis of centrosomal proteins in response to proteotoxic stress. (**A**) Schematic of ribosome profiling experiment. N = 4 samples per condition of control or ISR-null cells were treated with or without 100 nM bortezomib for 4 hr. Cells were lysed in the presence of 100 µg/mL cycloheximide and total mRNA and ribosome-protected fragments (RPFs) were sequenced to analyze translational efficiency (TE) genome-wide. (**B**) Filtering strategy to identify a high-confidence list of translationally regulated mRNAs targeted by the ISR following protein aggregate stress. To identify translationally regulated mRNAs, a two-step approach was taken. First, genes with RPF of a fold change >1.5 and padj <0.05 (DESeq2 generalized linear model) were identified. Second, within this list, genes with TE fold change >1.5 were selected (red dots, panels **C** and **D**). Finally, ISR target genes are identified by the overlap of mRNAs with significantly upregulated translation in two comparisons: (1) ISR-competent-bort vs. ISR-competent-vehicle and (2) ISR-competent-bort vs. ISR-null-bort. (**C**) Dot plot shows TE log₂ fold change vs. log₂ mean transcript abundance genome-wide in ISR-competent-bort vs. ISR-competent-vehicle highlights both upregulated genes as well as a downregulated group of

*Figure 3 continued on next page*

*Figure 3 continued*

highly abundant transcripts (see legend in panel **B**). As expected, highly expressed housekeeping transcripts are translationally downregulated. (**D**) Dot plot shows RPF log$_2$ fold change vs. total mRNA log$_2$ fold change genome-wide in ISR-competent-bort vs. ISR-null-bort (see legend in panel **B**). (**E**) Pie chart shows high-confidence translational upregulated ISR targets. Overlap between translationally upregulated genes in (1) ISR-competent-bort vs. ISR-competent-vehicle and (2) ISR-competent-bort vs. ISR-null-bort is shown. (**F**) Z-scored heatmap of TEs for 24 genes translationally induced by the ISR. Centrosomal proteins are highlighted in green lettering. (**G**) Top Gene Ontology (GO) terms identified in categories of biological processes and cellular components using translationally induced genes by the ISR. Centrosomal proteins are enriched among cellular components. #p=0.061, *p<0.05, **p<0.01, ***p<0.001 (Benjamini test). (**H**) Representative RPF tracks for *Atf5* reveal coding sequence translation are specifically induced in control cells treated with bortezomib. *Atf5* coding sequence is shaded in red. (**I**) Representative immunoblot confirms ISR-dependent induction of ATF4 and ATF5 following protein aggregate stress.

The online version of this article includes the following source data and figure supplement(s) for figure 3:

**Source data 1.** Original immunoblots show ATF5 and ATF4 level in S51 and S51A cells with or without proteotoxic stress (related to *Figure 3I*).

**Source data 2.** Raw read counts for ribosome profiling experiment.

**Figure supplement 1.** Quality control of ribosome profiling libraries.

**Figure supplement 2.** Ribosome occupancy changes occur in the absence of transcriptional changes.

of mRNA protected by ribosomes from RNase-I digestion (*McGlincy and Ingolia, 2017*; *Figure 3—figure supplement 1C*).

Having verified the efficacy of our data, we next focused on the translational response to proteotoxic stress in eIF2α-S51 control SCC cells bearing an intact ISR. To this end, we sought to identify mRNAs that displayed increased ribosome occupancy (RPF FC > 1.5 and padj(RPF) < 0.05) in bortezomib compared to vehicle-treated cells. To eliminate possible translational variances arising from transcriptional differences, we also normalized the RPF reads of each transcript according to total mRNA levels (*Ingolia et al., 2009*; *McGlincy and Ingolia, 2017*). This allowed us to identify genes with TEs (RPF/mRNA) that are sensitive to bortezomib, and whose translational changes are at the heart of ISR-mediated differences (*Figure 3B*). Interestingly, in our S51-SCC cells with an intact ISR, proteotoxic stress provoked the translational upregulation of 199 mRNAs (RPF FC > 1.5, padj (RPF) < 0.05, and TE FC > 1.5) (*Figure 3C*).

Next, we performed the same analysis to compare ISR-competent and ISR-null cells upon proteotoxic stress. This revealed 167 mRNAs whose translational upregulation was dependent on the presence of an intact ISR (S51-bort vs. S51A-bort RPF FC > 1.5, padj (RPF) < 0.05, and TE FC > 1.5) (*Figure 3D*). These findings clearly showed that when SCC cells are refractory to proteotoxic stress-induced phosphorylation of eIF2α, their translational program is selectively perturbed.

To curate a list of specific ISR-targeted mRNAs, we identified genes for which the translation changed in response to bortezomib and only in cells with an intact ISR. To this end, we generated a Venn diagram, comparing (1) the 199 mRNAs translationally upregulated in control cells following bortezomib treatment, and (2) the 167 mRNAs that failed to be translationally upregulated by bortezomib when the ISR was crippled. By this criterion, 24 mRNAs surfaced as candidates for bona fide ISR translational targets and showed strong enrichment across all four sets of translational replicates of proteotoxic stress-induced SCC cells harboring an intact ISR (*Figure 3E and F*, *Figure 3—figure supplement 2A*).

We used Gene Ontology enrichment analysis (GO-term) to ask if common biological processes or cellular components were translationally targeted by the ISR. As quality control, we first focused on the downregulated genes, resulting from the overlap of mRNA translationally downregulated upon bortezomib treatment in ISR-competent cells and that failed to be translationally downregulated in ISR-null cells. These 120 genes were clearly enriched in components of translational machinery, which is consistent with the known role of the ISR pathway in downregulating translation and housekeeping gene synthesis (*Figure 3—figure supplement 2B*).

Next, we performed GO-term analysis of the 24 upregulated ISR-targets (*Figure 3G*). Regulation of stress-induced transcription featured prominently in the top GO-terms for biological processes and was exemplified by the presence of ATF4, a well-established stress-induced transcriptional regulator that is activated in premalignant SCC cells by non-canonical translation when eIF2α is phosphorylated (*Sendoel et al., 2017*). Most intriguing, however, were the top three GO-terms for cellular components: centrosomal proteins, MTOC proteins, and microtubule cytoskeletal proteins (*Figure 3G* and genes highlighted in green in *Figure 3F*).

The centrosome is the major cytoplasmic MTOC within interphase cells (*Caviston and Holzbaur, 2006*; *Sanchez and Feldman, 2017*; *Woodruff et al., 2017*). In SCC cells, like most other mammalian cells, the MTOC is located near the nucleus, where it is surrounded by pericentriolar proteins important for plus-ended microtubule growth towards the cell periphery. The polarity of microtubules establishes polarized transport, which, depending upon the motor protein involved, can occur either towards the minus ends at the MTOC or along the plus ends of the microtubules towards the focal adhesions.

Although the ISR had not been previously found to regulate the microtubule cytoskeleton, we were intrigued by the possibility that the ISR was influencing microtubule dynamics in response to proteotoxic stress, which in turn might affect how efficiently proteins are cleared during the recovery phase. This possibility was all the more compelling because protein aggregate formation and clearance is known to depend on microtubule-dependent intracellular transport (*Kopito, 2000*). Of further intrigue was the mRNA for ATF5, which, like that for ATF4, displayed dramatic increases in translation in the absence of transcriptional upregulation and specifically in ISR-competent SCC cells exposed to proteotoxic stress (*Figure 3H and I*, *Figure 3—figure supplement 2C*). This tight translational control is likely favored by the presence of inhibitory uORFs in the 5'UTR of *Atf5* (*Figure 3H*; *Watatani et al., 2008*).

## The ISR protects centrosomal microtubule dynamics

Intriguingly, ATF5 has been previously demonstrated to play a non-transcriptional role at the centrosome (*Madarampalli et al., 2015*), strengthening the hypothesis that ISR activation drives a subset of centrosomal proteins to bolster the response to proteotoxic stress and excessive protein aggregation. We therefore set out to evaluate how the centrosome and microtubule dynamics were changing in response to stress in SCC cells with or without an intact ISR.

To evaluate the status of the centrosome during stress, we performed immunofluorescence for centrosomal markers pericentrin (PCNT) and γ-tubulin (TUBG1). Fluorescence intensities of both markers were significantly increased in control, but not ISR-null cells, and specifically during recovery from proteotoxic stress since no changes were observed in vehicle-treated cells or upon recovery from oxidative and ER stress (*Figure 4A*, *Figure 4—figure supplements 1 and 2A*). Quantifications further revealed that the overall size of the MTOC was enlarged specifically during recovery from proteotoxic stress and only when the ISR was intact (*Figure 4B and C*). Additionally, enlargement of the MTOC was dependent on eIF2α dephosphorylation, suggesting that while ISR-dependent synthesis of cytoskeletal proteins occurs early on during stress response, cellular recovery requires resolution of the ISR (*Figure 4—figure supplement 2B*).

While pre-oncogenic and fully transformed keratinocytes both significantly increased MTOC size during the recovery phase from proteotoxic stress, keratinocytes isolated from littermate control mice failed to do so (*Figure 4—figure supplement 3*). This is in line with the higher sensitivity to bortezomib of control cells (*Figure 1G*) and reinforces the idea that upon oncogenic transformation skin stem cells acquire an ISR-dependent pro-survival cellular program, which protects centrosomal dynamics.

The increased MTOC size was especially intriguing because the pericentriolar region that surrounds the MTOCs is a special site of protein catabolism within the cell, concentrating both autophagosome and lysosomes as well as proteasomes and proteins marked for degradation (*Freed et al., 1999*; *Liu et al., 2016*; *Wigley et al., 1999*). Moreover, although the highly transformed, long-passaged HeLa cell line lacks physiological relevance, it was notable that proteosome inhibition in these cells resulted in the accumulation of pericentriolar material, including pericentrin and γ-tubulin (*Didier et al., 2008*). In light of these data, we became intrigued by the possibility that the ISR may be at the crux of reinforcing microtubule dynamics following proteotoxic stress to promote cellular recovery.

If our premise was valid, microtubule-dependent processes should be vulnerable when cells cannot trigger an ISR in the face of stressful situations. To evaluate microtubule dynamics during the recovery phase of our cells following proteotoxic stress exposure, we briefly interrupted existing microtubule assembly/disassembly dynamics with nocodazole and then examined nascent microtubule nucleation initiated from the centrosome (*Figure 4D and E*, *Figure 4—figure supplement 4A*).

In the absence of stress, the ISR-null state was not associated with differences in MTOC-initiated microtubule dynamics (*Figure 4E*). In striking contrast, the vulnerability of microtubule dynamics in the ISR-null state became clear when we exposed our cells to proteotoxic stress and monitored the

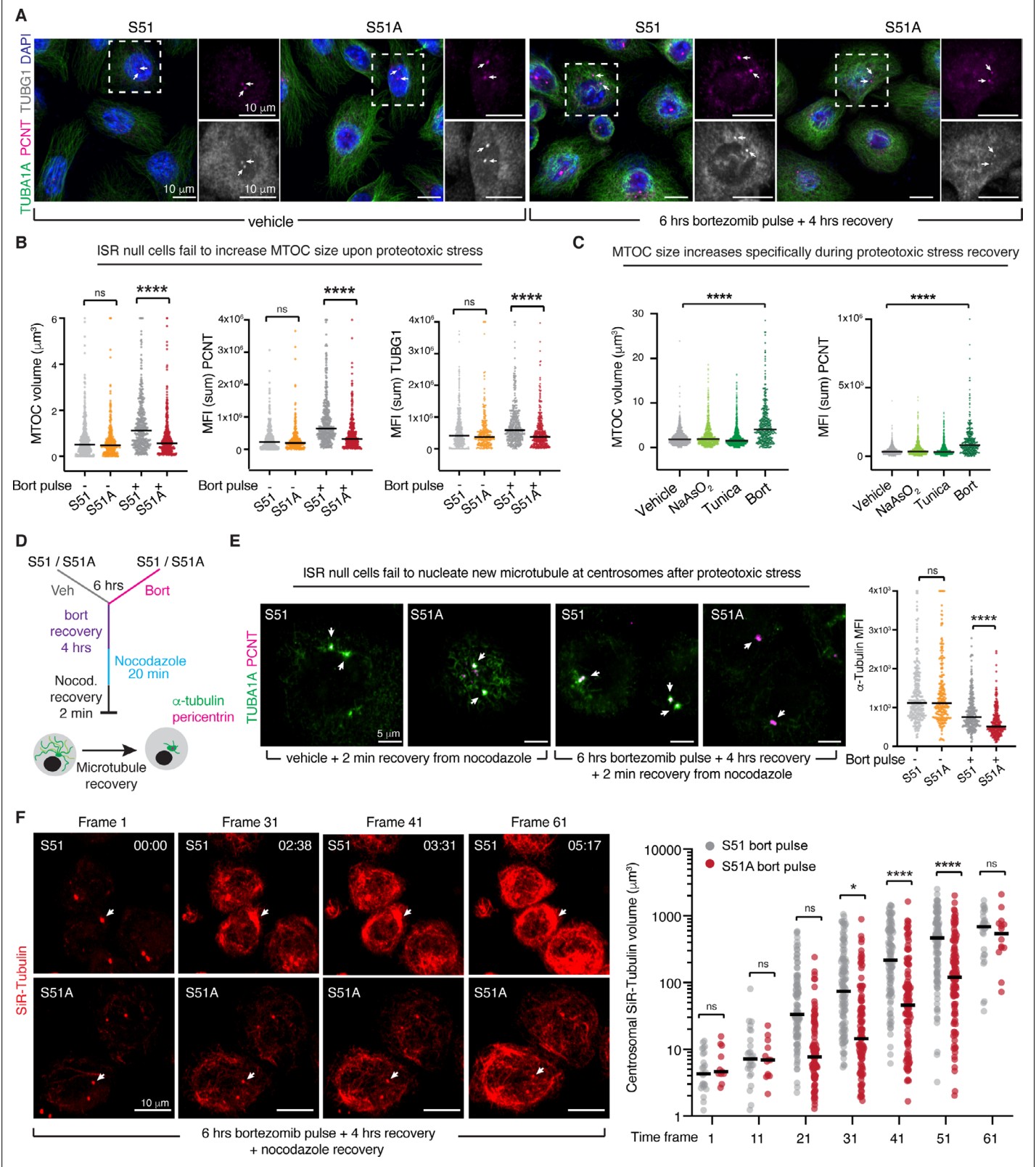

**Figure 4.** The integrated stress response (ISR) protects centrosomal microtubule dynamics during recovery from protein aggregate stress. (**A, B**) Representative immunofluorescence images of S51 and S51A cells vehicle-treated or treated with a bortezomib pulse (100 nM, 6 hr) and let to recover for 4 hr. Immunolabeling is to highlight the centrosomal microtubule-organizing center (MTOC) (pericentrin, PCNT, and γ-tubulin, TUBG1), and the cytoplasmic microtubules (α-tubulin, TUBA1A). Scale bar 10 μm. White arrows indicate centrosomal MTOCs. (**B**) Both volume and mean sum

*Figure 4 continued on next page*

*Figure 4 continued*

fluorescence intensity of MTOC markers are increased specifically in control cells recovering from protein aggregate stress. All data are visualized on a per centrosome basis from N = 2 independent experiments. ****p<0.0001, ns, no statistical significance (two-way ANOVA with multiple comparisons). (**C**) S51 squamous cell carcinoma (SCC) cells were treated with either vehicle, 50 μM sodium arsenite, 100 ng/μL tunicamycin, or 100 nM bortezomib for 6 hr, followed by a recovery period of 4 hr. MTOC volume and mean sum fluorescence intensity are plotted for two independent biological experiments. ****p<0.0001 (two-way ANOVA with multiple comparisons). (**D**) Schematic of microtubule recovery assay to assess the capacity of centrosomes to nucleate new microtubules following 4 hr recovery from protein aggregate stress. Cells with or without protein aggregates are treated with 13 μM nocodazole for 20 min to depolymerize microtubules. Nocodazole is washed from the media, and new microtubules are allowed to form during a 2 min incubation at room temperature. Cells are fixed and MTOCs (PCNT+) and cytoplasmic microtubules (TUBA1A) are quantified and analyzed by immunofluorescence. (**E**) Representative images of centrosomal microtubule nucleation following nocodazole wash out in control and ISR-null cells with or without protein aggregates. Centrosomal microtubule nucleation is analyzed by measuring α-tubulin signal at the centrosome, which is defined as the pericentrin-positive volume. White arrows indicate centrosomal MTOCs. Scale bar 5 μm. Quantification of mean centrosomal α-tubulin intensity indicates that control cells are partially protected against impaired microtubule dynamics during stress. N = 2 independent experiments. ****p<0.0001, ns, no statistical significance (two-way ANOVA with multiple comparisons). (**F**) Still images from live-imaging experiments show time-dependent growth of microtubule at the centrosome following nocodazole removal. S51 and S51A SCC cells were treated with a bortezomib pulse and let to recover for 4 hr prior to nocodazole treatment. Quantification on the right shows volume of SiR-tubulin-labeled microtubules growing from the centrosome. Each dot represents one cell. Data from two independent biological experiments are plotted. *p<0.05, ****p<0.0001, ns, no statistical significance (two-way ANOVA with multiple comparisons). Scale bar 10 μm.

The online version of this article includes the following video and figure supplement(s) for figure 4:

**Figure supplement 1.** Increased centrosomal microtubule-organizing center (MTOC) size upon proteotoxic stress is dependent on the integrated stress response (ISR).

**Figure supplement 2.** Increased centrosomal microtubule-organizing center (MTOC) size is restricted to proteotoxic stress and depends on eIF2α dephosphorylation.

**Figure supplement 3.** Increased centrosomal microtubule-organizing center (MTOC) size is unique to transformed keratinocytes.

**Figure supplement 4.** Characterization of nocodazole recovery experiment.

**Figure 4—video 1.** Microtubule growth is not affected in integrated stress response (ISR)-null cells at steady state.
https://elifesciences.org/articles/77780/figures#fig4video1

**Figure 4—video 2.** Microtubule growth from the centrosome is delayed in integrated stress response (ISR)-null cells recovering from proteotoxic stress.
https://elifesciences.org/articles/77780/figures#fig4video2

recovery process. Even after 4 hr of bortezomib recovery, microtubule growth from the MTOC was still markedly impaired in the ISR-null compared to control SCC cells (*Figure 4E*).

To better track microtubule dynamics, we treated S51 and S51A cells with SiR-tubulin, a cell-permeable, fluorescent dye based on the microtubule binding drug docetaxel. Live imaging of S51 and S51A cells showed no overt differences in the microtubule cytoskeleton of unstressed cells. However, when combined with nocodazole recovery, SiR-tubulin imaging revealed a significant delay in new microtubules nucleating from the centrosomes in S51A cells specifically during recovery from proteotoxic stress (*Figure 4F*, *Figure 4—figure supplement 4B*, *Figure 4—video 1*, *Figure 4—video 2*). This is in agreement with our data showing that, while ISR-null SCC cells fare considerably worse than ISR-competent SCC cells in response to proteotoxic stress, they eventually recover (*Figure 2D*). Intriguingly, we also noticed that while the microtubules in ISR-competent SCC cells were mostly nucleating from the centrosomes, a majority of ISR-null SCC cells showed prominent non-centrosomal microtubule growth (*Figure 4—video 2*) suggestive of a compensation mechanism to allow for cellular recovery. Overall, since microtubule dynamics were independent of the ISR in the unstressed state, these results suggested that the ISR is required to reinforce the microtubule dynamics involved in cellular recovery following proteotoxic stress.

## The ISR is required for aggresome formation

Our evidence that the ISR functions in protecting microtubule dynamics during proteotoxic stress was all the more compelling because of known role of microtubules in assembling ubiquitinated protein aggregates into larger membraneless structures called aggresomes (*Johnston et al., 1998*; *Wigley et al., 1999*). Through their ability to associate with proteasomes and autophagosomes, aggresomes appear to be a key intermediate in the clearance of improperly folded proteins.

With these insights in mind, and the general view that aggresomes function beneficially in cellular recovery from proteotoxic stress, we evaluated aggresome formation in our SCC cells following

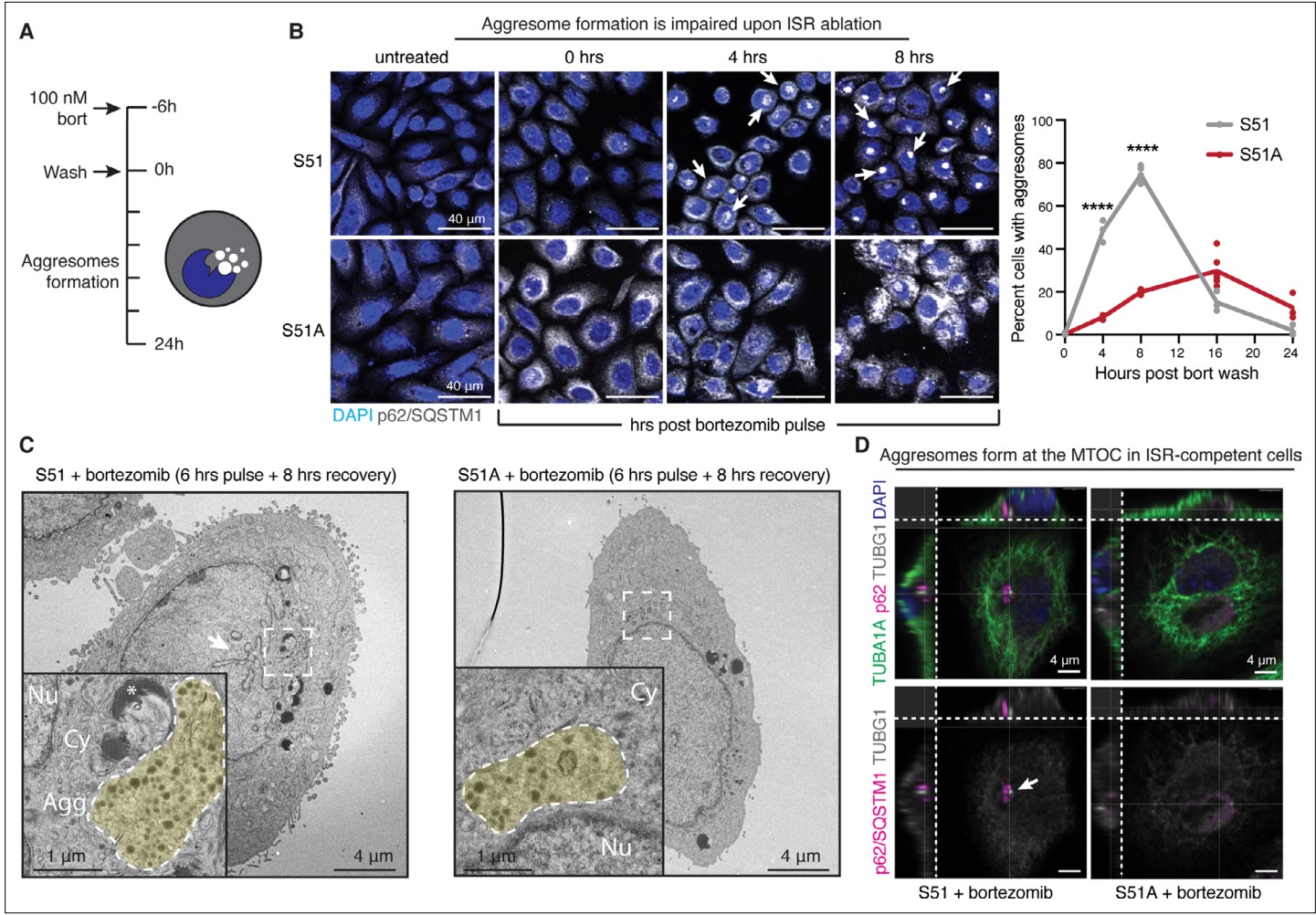

**Figure 5.** The integrated stress response (ISR) is required for aggresome formation. (**A**) Schematic of aggresome formation assay. During recovery from proteotoxic stress, aggresome formation is assessed by anti-p62/SQSTM1 immunofluorescence, which appears as a well-defined, perinuclear puncta. (**B**) Time course of aggresome formation in control and ISR-null cells indicates diminished aggresome formation in cells with an ablated ISR. White arrows indicate aggresomes. Quantification of the percentage of aggresome-positive cells. N = 2 independent experiments with at least 145 cells quantified per condition. Error bars represent 95% confidence interval calculated by Wilson's method for confidence interval of a proportion. ****p<0.001 using two-sample Z-test for a proportion (comparisons: control vs. ISR-null at 4 and 8 hr). Scale bar 40 μm. (**C**) Aggresomes visualized at the ultrastructural level by transmission electron microscopy. In control cells, aggresomes characteristically appear as a perinuclear cluster of electron-dense granules and partially degraded organelles that often deform the nuclear membrane. Fewer aggresomes form in ISR-null cells, and those that do are smaller, more diffuse and rarely indent the nucleus. White arrow indicates nuclear envelope deformation. Scale bar 4 μm. Yellow shade in the zoomed-in images at the lower left highlights aggresomes and protein aggregates. Nu, nucleus. Cy, cytoplasm, Agg, aggresome. *Lipid droplet. Scale bar 1 μm. (**D**) Confocal immunofluorescence microscopy and 3D reconstruction to visualize the location of the aggresome in relation to the microtubule-organizing center (MTOC) (γ-tubulin, TUBG1) and cytoplasmic microtubules (α-tubulin, TUBA1A). In control cells, aggresomes appear as dense p62-positive structures directly adjacent to the MTOC puncta. Representative cells from 8 hr post-bortezomib washout are shown. White arrow indicates MTOC. Scale bar 4 μm.

The online version of this article includes the following figure supplement(s) for figure 5:

**Figure supplement 1.** PKR-like endoplasmic reticulum kinase (PERK) inhibition inhibits aggresome formation independently from the integrated stress response (ISR).

**Figure supplement 2.** The integrated stress response (ISR) promotes aggresome formation upon recovery from proteotoxic stress.

**Figure supplement 3.** Aggresomes accumulate at the centrosome upon recovery from proteotoxic stress.

**Figure supplement 4.** Aggresome formation is dependent on microtubule dynamics.

withdrawal of bortezomib (*Figure 5A*). To this end, we used immunofluorescence for p62/SQSTM1, a protein involved in shuttling ubiquitinated protein aggregates to the aggresome (*Christian et al., 2010*). As judged by convergence of intense anti-p62 immunofluorescence to a single large perinuclear spot within each S51 cell, misfolded protein aggregates began to coalesce into the aggresome soon after bortezomib withdrawal, peaking at approximately 8 hr into the recovery phase (*Figure 5B*, top panels). By 24 hr, the recovery phase appeared to be complete as the aggresome was no longer present (see quantifications at right). These findings were in good agreement with the clearance of ubiquitinated misfolded proteins that had accumulated during the proteosome block (*Figure 2E*).

In striking contrast to S51 cells, the S51A SCC cells lacking an intact ISR showed pronounced defects in their ability to clear misfolded proteins that had accumulated during bortezomib treatment. The rise in cytoplasmic p62 immunofluorescence during the proteosomal block indicated that small protein aggregates had formed in the S51A cells exposed to proteotoxic stress (*Figure 5B*, bottom panels). However, even at 8 hr after bortezomib withdrawal, only a few S51A cells displayed the bright perinuclear spot reflective of the aggresome. Moreover, while S51 SCC cells had cleared their protein aggregates by 24 hr after culturing in normal media, S51A SCC cells still exhibited anti-p62 coalescence, indicative of a marked delay in clearance of unfolded proteins. In agreement with our previous data that PERK signaling contributed to clearance of ubiquitinated proteins independently of the ISR (*Figure 2—figure supplement 3D*), PERK inhibition resulted in a significant reduction in the percentage of cells forming aggresomes in both ISR-competent and ISR-null SCC cells (*Figure 5—figure supplement 1*).

Further signs of cellular defects during the recovery process were evident at the ultrastructural level. After 8 hr of recovery, electron-dense protein aggregates pushing into the nucleus were observed in the perinuclear regions of S51 but not S51A SCC cells. These structures were absent in unstressed SCC cells. (*Figure 5C*, *Figure 5—figure supplement 2A–C*). Consistent with the notion that the aggresomes are trafficked via the microtubules, further analysis of our electron microscopy images revealed accumulation of membrane-less structure (i.e., aggresomes) at the centrosomes in S51 SCC cells during recovery from proteotoxic stress (*Figure 5—figure supplement 3A and B*). Co-immunolabeling for p62 and γ-tubulin confirmed that the perinuclear structures present in S51 cells were indeed aggresomes (*Figure 5D*). Moreover, S51 cells displayed signs of nuclear deformation, often observed during the aggresome clearance phase. By contrast, when detected in S51A cells, aggresomes tended to be small and less distinct. Moreover, S51A cells often displayed widespread vacuole-like structures, which have been reported to occur when the ER becomes overwhelmed with misfolded proteins (*Mimnaugh et al., 2006*; *Figure 5—figure supplement 2C*).

The surprising dependence upon an intact ISR for SCC cells to form aggresomes during proteotoxic stress recovery led us to hypothesize that the ISR might function critically in sustaining the necessary microtubule dynamics that enables efficient transport of misfolded proteins to the MTOC, where they can then form aggresomes and be targeted for perinuclear proteosomal and autophagosomal clearance. To further challenge this relationship between microtubule dynamics and aggresome formation, we used the drug, paclitaxel, a chemotherapy that stabilizes microtubules while at the same time disrupts their dynamics. In line with our hypothesis, microtubule stabilization with paclitaxel potently blocked aggresome formation, phenocopying the ISR-null cells during this process (*Figure 5—figure supplement 4A*). Blocking microtubules polymerization by treating cells with nocodazole had a similar effect and resulted in a significant reduction in aggresome formation (*Figure 5—figure supplement 4B*).

## The ISR is required for migration and focal adhesion homeostasis following protein aggregate stress

During the course of our previous experiments, we made several observations regarding the cellular morphology, which supported the conclusion that the ISR was required to maintain proper microtubule dynamics in the face of protein aggregate stress. First, we noticed that control, but not ISR-null, cells went through a dramatic change in cell shape while recovering from aggregate stress. Specifically, control cells 'unspread,' becoming more compact and tall between 4 and 8 hr after washing out bortezomib, the timepoints correlating with peak aggresome formation. ISR-null cells, on the other hand, did not similarly round, and instead they maintained a flattened shape with some cells also forming elongated processes. Examples of these differences are shown in *Figure 6—figure supplement 1*.

The long cellular extensions were reminiscent of that seen when keratinocytes displayed defects in the turnover of focal adhesions, a process that we had previously shown depends not only upon focal adhesion proteins (*Schober et al., 2007*), but also on the ability of microtubules to deliver turnover cargo to the focal adhesions (*Wu et al., 2008*). Specifically, microtubule-mediated cargo transport is essential for focal adhesion disassembly and cell migration (*Ezratty et al., 2005*; *Yue et al., 2014*). While the transport of protein aggregates to the centrosome involves minus end-directed molecular motors, transport to the focal adhesions requires plus end-directed motors. Reasoning that a defect in MTOC-mediated microtubule dynamics might affect both processes, we examined the focal adhesions in control and ISR-null cells following proteotoxic stress. As judged by immunofluorescence for vinculin, upon proteotoxic stress the ISR promoted a remodeling in cell morphology evident by diminished focal adhesion and F-actin staining. By contrast, ISR-null cells retained large focal adhesions suggestive of impaired cytoskeletal dynamics (*Figure 6A*; *Geiger et al., 1984*; *Wu et al., 2008*). This ISR-mediated difference was specific to stress and was not seen when control and ISR-null cells were cultured in normal media under unstressed conditions. By looking back at our ribosome profiling dataset, we found a significant enrichment of focal adhesion genes within mRNAs with reduced ribosome occupancy upon ISR induction (*Figure 3—figure supplement 2B*). While further analysis will be needed to evaluate protein levels of focal adhesion components in the absence of an ISR, these data suggest that ISR-mediated translational control stands at the crux of these dramatic morphological changes by selectively upregulating microtubule dynamics and, at the same time, downregulating focal adhesion components.

To directly evaluate focal adhesion dynamics in our cells, we generated a zyxin-iRFP construct that would allow for live imaging of cells. Indeed, ISR-null SCC cells recovering from proteotoxic stress possessed large, stable focal adhesions that did not disassemble, rendering the cell immobile over the time frame, whereas control SCC cells disassembled their large focal adhesions and displayed considerable dynamics (*Figure 6B*, *Figure 6—video 1*). To confirm these aberrant dynamics, we performed a scratch wound assay to induce cell migration following bortezomib treatment. In agreement with our previous data showing large and stable focal adhesion, ISR-null cells showed markedly impaired mobility relative to control cells and failed to migrate into the wound site (*Figure 6C*, *Figure 6—video 2*, *Figure 6—video 3*). Taken together, these results provided further evidence that in the absence of an ISR, SCC cells exposed to proteotoxic stress are slow to ignite the microtubule dynamics necessary for recovery.

## ATF5 acts downstream of the ISR to promote aggresome formation and recovery from proteotoxic stress

Our collective data strongly implicated the ISR in regulating microtubule dynamics with the purpose of promoting aggresome assembly and the efficient clearance of protein aggregates that accumulate during proteotoxic stress. Additionally, our analysis of ISR-dependent translational changes revealed a selective upregulation of centrosomal proteins upon proteotoxic stress (*Figure 3G*). In particular, we were intrigued by ATF5, which has been described as an integral component of the MTOC (*Madarampalli et al., 2015*). Strikingly, in contrast to ATF4 that is broadly induced to stress, we found that ATF5 is translated only during proteotoxic stress (*Figure 7A*). In agreement with the notion that ATF5 may play a selective role in clearance of protein aggregates, confocal microscopy revealed that upon recovery from proteotoxic stress ATF5 localized at the centrosome in ISR-competent SCC cells (*Figure 7B*). Hence, we reasoned that translational upregulation of ATF5 might be needed to regulate microtubule dynamics in the face of proteotoxic stress. To test this hypothesis, we engineered ISR-null cells to overexpress doxycycline-inducible, GFP-tagged ATF5 (*Figure 7C*, *Figure 7—figure supplement 1A*).

To probe whether ATF5 is needed to clear protein aggregates, we first examined the clearance of ubiquitinated proteins during the recovery phase following a pulse of bortezomib. Remarkably, expression of ATF5 significantly improved the ability of ISR-null cells to clear ubiquitinated proteins, as assessed by immunoblot of ubiquitinated proteins in the insoluble fraction of cells after 24 hr recovery from bortezomib (*Figure 7D and E*). Strikingly, this corresponded with a significant increase in aggresome formation, indicating that ISR-mediated ATF5 upregulation is needed to promote the retrograde transport of misfolded proteins and the subsequent accumulation of protein aggregates at the MTOC (*Figure 7F*). This effect was mediated via protection of microtubule dynamics as evidenced

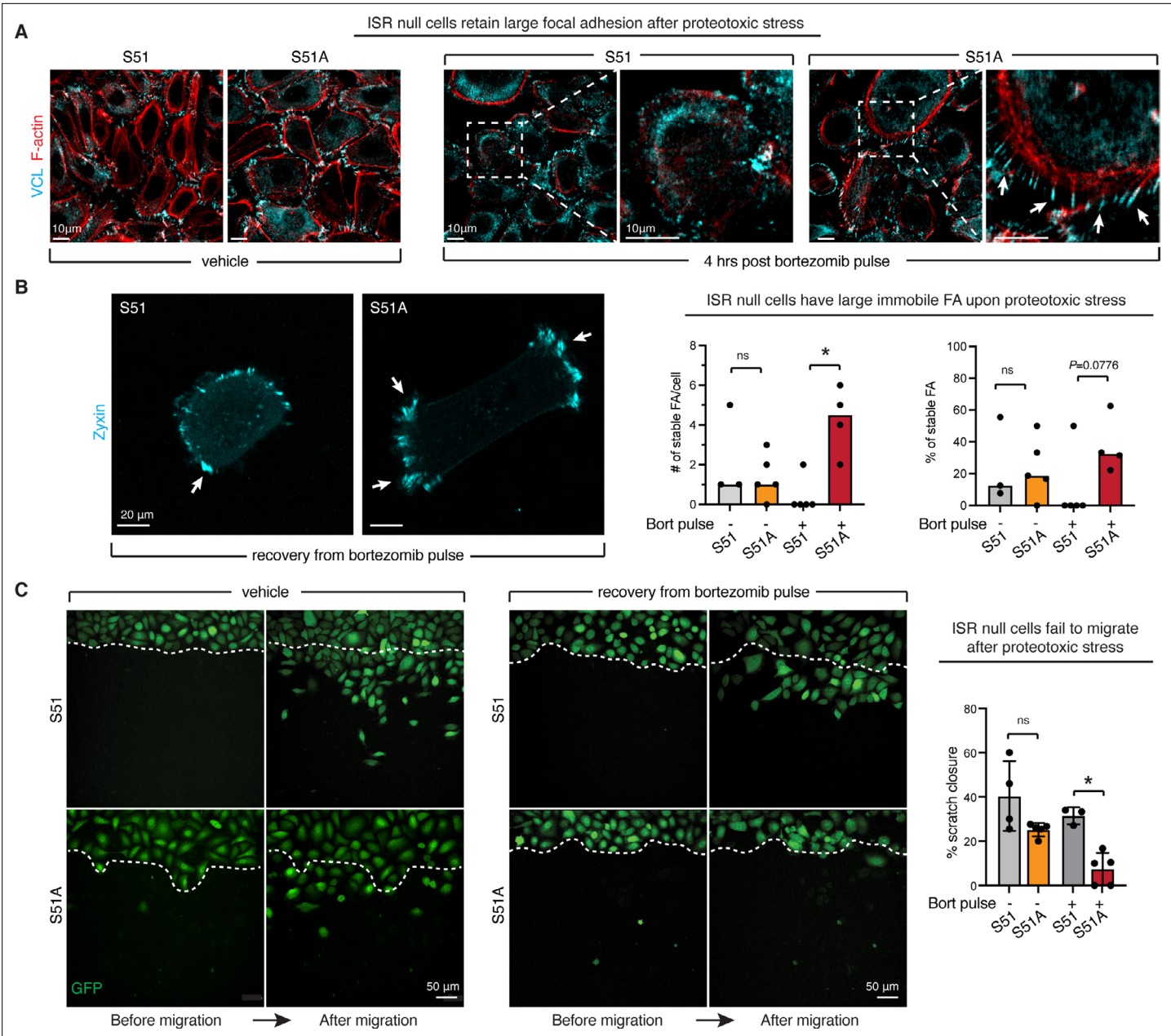

**Figure 6.** The integrated stress response (ISR) is required for focal adhesion remodeling and cell migration upon proteotoxic stress. (**A**) Impaired focal adhesion dynamics in ISR-null cells. Immunofluorescence for vinculin and F-actin shows that during recovery from protein aggregate stress the ISR promotes the remodeling of focal adhesions. White arrows show characteristic long cellular extensions connected to enlarged focal adhesions in the ISR-null but not control cells recovering from protein aggregate stress. Consistent with the known role of microtubules in targeting focal adhesion turnover, these features resemble those of keratinocytes lacking microtubule-focal adhesion targeting proteins (**Wu et al., 2008**). Scale bar 10 µm. (**B**). Static frames from live imaging of zyxin-iRFP cells. White arrows show large (10 µm³ or greater) and stable (over the whole duration of imaging period) focal adhesion clusters. Bar graphs show total number of stable and large focal adhesion clusters per cell (left) and percentage of stable over total focal adhesion clusters (right) over a 14 min live-imaging time course. 3–5 cells were imaged per condition in two independent biological replicates. *p<0.05, ns, no statistical significance (*t*-test). (**C**) Live imaging of GFP-positive cells migrating in a scratch assay reveals loss of migratory capacity in ISR-null cells recovering from protein aggregate stress (6 hr of 100 nM bortezomib followed by wash). Scratch closure is quantified during an 8 hr interval, in N = 2 independent experiments with n = 3–5 technical replicates per condition. Bar graph shows percentage of scratch closure ± SD. *p<0.05, ns, no statistical significance (one-way ANOVA with multiple comparisons).

The online version of this article includes the following video and figure supplement(s) for figure 6:

**Figure supplement 1.** The integrated stress response (ISR) promotes dramatic morphological changes in response to proteotoxic stress.

*Figure 6 continued on next page*

*Figure 6 continued*

**Figure 6—video 1.** Focal adhesion dynamics in integrated stress response (ISR)-competent (S51) or ISR-null S51A squamous cell carcinoma (SCC) cells in untreated conditions or upon recovery from proteotoxic stress.

https://elifesciences.org/articles/77780/figures#fig6video1

**Figure 6—video 2.** Scratch assay in integrated stress response (ISR)-competent (S51) squamous cell carcinoma (SCC) cells in untreated conditions or upon recovery from proteotoxic stress.

https://elifesciences.org/articles/77780/figures#fig6video2

**Figure 6—video 3.** Scratch assay in integrated stress response (ISR)-null (S51A) squamous cell carcinoma (SCC) cells in untreated conditions or upon recovery from proteotoxic stress.

https://elifesciences.org/articles/77780/figures#fig6video3

by increased pericentrin mean fluorescence intensity at the centrosome in ISR-null cells expressing ATF5 after recovery from bortezomib treatment (*Figure 7G*). Strengthening this mechanistic link, short-hairpin-RNA-mediated ATF5 knockdown in ISR-competent cells resulted in higher accumulation of ubiquitinated proteins and reduced aggresome formation during recovery from proteotoxic stress (*Figure 7—figure supplement 1B-E*).

Building on our previous findings that normal keratinocytes were more sensitive to bortezomib compared to their transformed counterpart and failed to protect MTOC dynamics during the recovery phase from proteotoxic stress (*Figure 1G*, *Figure 4—figure supplement 3*), we asked whether ATF5 induction occurred only in the transformed state. In response to bortezomib, control keratinocytes mounted an ISR, as judged by eIF2α phosphorylation, and upregulated ATF4 although at lower level compared to transformed cells. Yet, despite expressing similar levels of *Atf5* mRNA, untransformed cells failed to induce *Atf5* translation and showed no ATF5 protein expression (*Figure 7H*, *Figure 7—figure supplement 1F*). These findings reinforce the notion that upon transformation SCC stem cells alter their ISR program to more efficiently recover from proteotoxic stress (*Figure 7H*, *Figure 7—figure supplement 1F*).

Finally, we asked whether ISR-mediated ATF5 induction promotes cell survival upon proteotoxic stress of SCC cells in vivo. To this end, we turned to our grafting model where ISR-null cells displayed increased sensitivity to proteasome inhibition. Consistent with our hypothesis, ATF5 expression rescued the selective sensitivity to bortezomib shown by ISR-null cells. This occurred in the absence of changes in viability in tumors at steady state (*Figure 7I*). Altogether, these data implicate ATF5 as a critical ISR target in SCC stem cells, which orchestrates microtubule dynamics to facilitate the clearance of protein aggregates and promote cell recovery upon proteotoxic stress.

## Discussion

Although a handful of reports have linked the actin cytoskeleton to eIF2α dephosphorylation (*Chambers et al., 2015*; *Chen et al., 2015*), cytoskeletal regulation has not been viewed as a primary function of the ISR. In this study, we unearthed a novel and essential role for the ISR in regulating the cytoskeleton, specifically through preserving microtubule dynamics in stressful situations. This mechanism appears to be unique to proteotoxic stress and is required for cells to accomplish the necessary intracellular trafficking to efficiently clear misfolded proteins and prevent them from accumulating and overtaxing the ER.

Our findings led us to the remarkable conclusion that the ISR is not only required to mount a stress response, but also to instruct cell function during recovery from stress. Although the existence of negative feedback loops that downregulate the ISR upon termination of stress has long been appreciated (*Novoa et al., 2001*), our data more clearly elucidate a specific role of the ISR during cellular recovery.

By temporally profiling the translational differences that arise when proteotoxically stressed SCC cells are unable to mount an ISR, we gained insights into the mechanisms underlying the ISR's importance. Specifically, we learned that when ISR-competent cells are exposed to proteotoxic stress, they redirect their translational machinery to a cohort of mRNAs encoding centrosomal proteins. We show that these proteins function in bolstering the MTOC and reinforcing its microtubule dynamics. This then facilitates efficient microtubule-mediated transport of misfolded proteins to the perinuclear space, where they can be assembled into aggresomes and targeted for ubiquitin-mediated

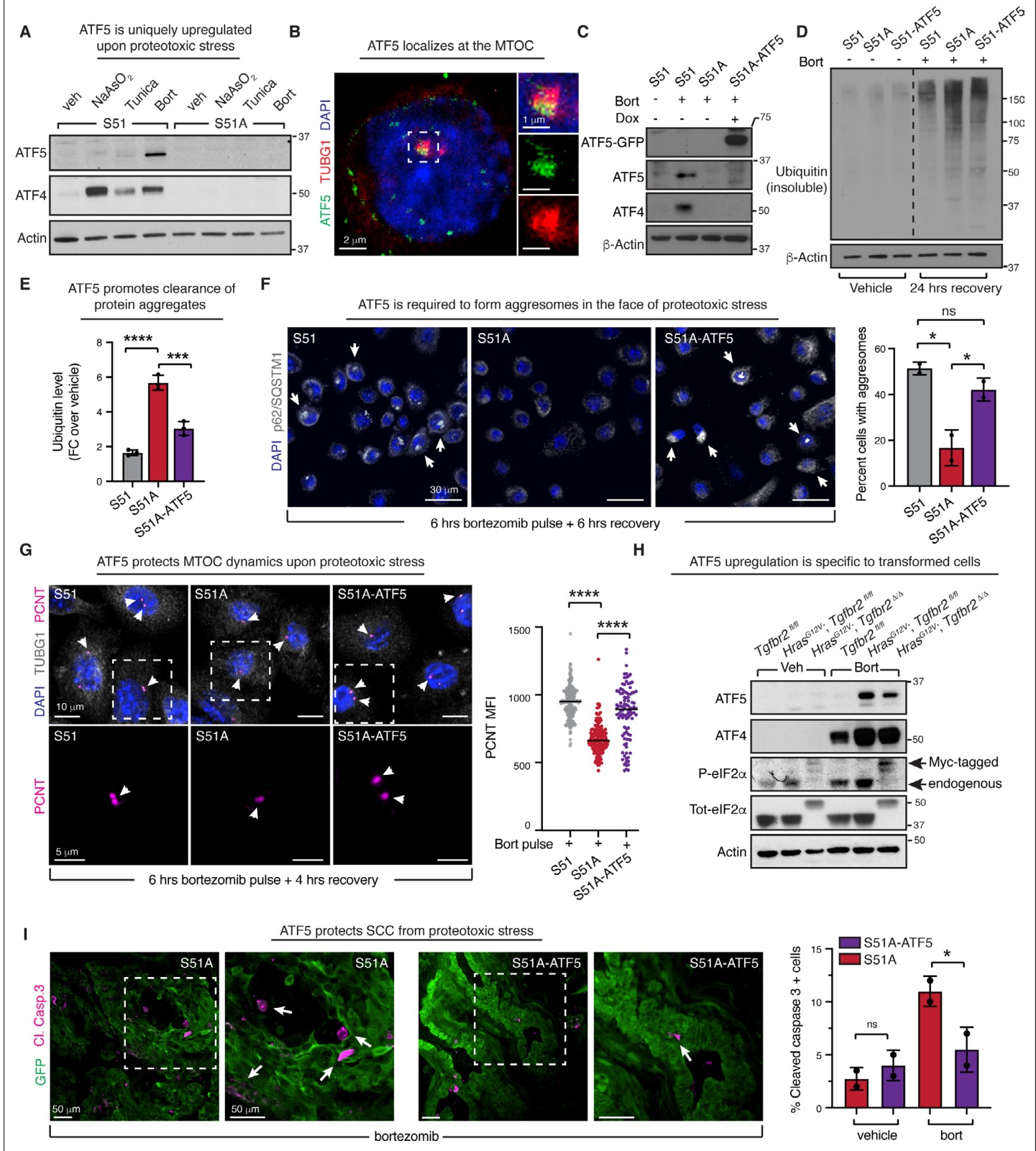

**Figure 7.** Integrated stress response (ISR)-induced ATF5 protects microtubule dynamics and promotes clearance of protein aggregates. (**A**) ATF5 is selectively upregulated upon proteotoxic stress in an ISR-dependent fashion. S51 and S51A were treated with either vehicle, 50 μM sodium arsenite, 100 ng/μL tunicamycin, or 100 nM bortezomib for 6 hr. Representative blots from three independent experiment are shown. (**B**) Confocal microscopy confirms previously reported ATF5 localization at the microtubule-organizing center (MTOC). Insets on the right show higher magnification images with

*Figure 7 continued on next page*

*Figure 7 continued*

merged (top) and ATF5 (middle) and γ-tubulin (bottom) separately. (**C**) Immunoblot confirms ATF5 expression upon Dox administration (1 μg/mL for 24 hr) in proteotoxically stressed S51A-ATF5 cells where endogenous ATF5 translation is suppressed. (**D**) ATF5 expression partially rescues the ability of proteotoxically stressed ISR-null cells to clear ubiquitinated proteins. Representative immunoblot (of three independent biological replicates) shows level of ubiquitinated proteins in the radio-immunoprecipitation assay (RIPA)-insoluble fraction of S51, S51A, and S51A-ATF5 squamous cell carcinoma [SCC] cells in the absence or proteotoxic stress or following 24 hr recovery from a 6 hr bortezomib pulse. (**E**) Quantifications of experiments in (**D**). Bar graph shows ubiquitin levels ± SD at 24 hr recovery normalized over actin and relative to ubiquitin levels in untreated cells in three independent biological replicates. *$p<0.05$, ns, no statistical significance (one-way ANOVA with multiple comparisons). (**F**) ATF5 promotes aggresome formation. Aggresomes are visualized by p62/SQSTM1 immunofluorescence in SCC cells treated with a 6 hr bortezomib pulse followed by a 4 hr recovery period. White arrows point to aggresomes. Scale bar 30 μm. Bar graph shows percentage of cells with aggresomes ± SD in two independent biological replicates. Approximately 300 cells were quantified per each experiment. *$p<0.05$, ns, no statistical significance (one-way ANOVA with multiple comparisons). (**G**) ATF5 protects centrosomal MTOC dynamics. Centrosomal MTOC is visualized by pericentrin and γ-tubulin immunofluorescence. Cells were treated with bortezomib for 6 hr and let to recover for 4 hr. Scale bar 10 μm. Bottom row shows higher magnification of the area highlighted in top row. White arrows indicate MTOCs. Scale bar 5 μm. Bar graph shows mean fluorescence intensity for pericentrin in two independent biological replicates. ****$p<0.0001$ (one-way ANOVA with multiple comparisons). (**H**) Representative immunoblot shows selective upregulation of ATF5 upon bortezomib in transformed keratinocytes. Cells were treated with either vehicle or 100 nM bortezomib for 6 hr. Representative blot of three independent experiment is shown. (**I**) ATF5 induction protects ISR-null cells from protein aggregates-induced cytotoxicity. Immunofluorescence shows SCC tumors (GFP+) after treatment with bortezomib (1.2 mg/kg i.p.). Cell death is quantified by cleaved caspase 3-positive cells. White arrows point to cleaved caspase 3-positive cells. Scale bar 50 μm. Bar graph shows percentage of cleaved caspase 3-positive cells per tumor ± SD in two independent biological replicates. *$p<0.05$, ns, no statistical significance (one-way ANOVA with multiple comparisons).

The online version of this article includes the following source data and figure supplement(s) for figure 7:

**Source data 1.** Original immunoblots show ATF5 and ATF4 level in S51 and S51A cells upon different stresses (related to *Figure 7A*).

**Source data 2.** Original immunoblots show ATF5 and ATF4 level in S51A cells overexpressing exogenous ATF5 (related to *Figure 7C*).

**Source data 3.** Original immunoblots show ubiquitin levels in S51A cells overexpressing exogenous ATF5 (related to *Figure 7D*).

**Source data 4.** Original immunoblots show ATF5 and ATF4 level in indicated genotypes with and without proteotoxic stress (related to *Figure 7H*).

**Source data 5.** Raw numerical data for all graphical representation (related to *Figures 1–7*).

**Figure supplement 1.** ATF5 is needed for efficient protein aggregate clearance.

degradation during the stress recovery phase. Indeed, as we showed, when eIF2α cannot be phosphorylated and the ISR core is thereby crippled in proteotoxically stressed cells, the centrosomal proteins are not translated, and microtubule dynamics emanating from the MTOC are disrupted. As judged by deficiencies in the aggresome assembly and in focal adhesion turnover, both retrograde and anterior grade transport of microtubule cargo are slow to recover following stress, thereby impairing cell fitness. Compellingly, the ISR not only upregulates a subset of centrosomal proteins, but also downregulates focal adhesion components, surfacing a network of ISR-regulated cytoskeletal dynamics that directs morphological changes critical for cell recovery.

Microtubule trafficking is required for misfolded proteins to accumulate at the perinuclear space, and our results provided compelling evidence that the ISR functions in this process. However, our findings also pointed to a hitherto unappreciated role of the ISR in aggresome assembly specifically. Our studies revealed that this function is in part mediated by ATF5, a translational target of the ISR in proteotoxically stressed SCC cells and a previously documented structural component of the centrosomal MTOC (*Madarampalli et al., 2015*). We showed that during recovery from protein aggregate stress the MTOC size increases concomitantly with aggresome formation, and both of these events depend upon an intact ISR. Our data support a model in which ISR-mediated translational upregulation of centrosomal proteins, including ATF5, is required to remodel the MTOC and concentrate protein aggregates in the perinuclear space so that they can be degraded and cleared by proteosomes and autophagosomes.

Our findings that *Atf5* translational upregulation is limited to proteotoxic stress and unique to transformed skin stem cells is particularly intriguing. *Atf5* possesses uORFs in its 5′UTR that are known to regulate its translation, but it remains unclear which factors modulate their translation and promote synthesis of ATF5 protein. In this regard, it is curious that while bortezomib induced the lowest phosphorylation of eIF2α, it uniquely induced *Atf5* translation. Together, these findings point to a view of the ISR as a dynamic program that can be finely tuned in the face of distinct stresses and cellular states. Finally, these data highlight a cancer-specific mechanism of recovery from proteotoxic stress, supporting the idea that the ISR contributes to therapy resistance in SCC.

Several studies have provided tantalizing evidence that some cells may be able to asymmetrically partition their centrosomal aggregates when division resumes following proteotoxic stress, such that one daughter (e.g., a stem cell) remains healthy, while the other daughter inherits the aggregates and becomes slated to differentiate (*Morrow et al., 2020*; *Rujano et al., 2006*; *Singhvi and Garriga, 2009*). We did not see evidence for this in our SCC cells. Rather, there was a lag in cell cycle reentry such that proliferation began after aggregates were cleared, and this coupling was maintained in ISR-null cells (*Figures 2D and 5B*). This coupling was especially striking in comparing the behaviors of ISR-null vs. ISR-competent cells. Thus, during the recovery phase, ISR-null cells were delayed by >24 hr in both aggregate clearance and cell cycle re-entry. That said, when the ISR was crippled either in vivo or in vitro, SCC cells were jeopardized in their overall ability to survive proteotoxic stress. In this scenario, as a major regulator of cell survival in response to proteotoxic stress, our findings suggest that by coupling pharmacological inhibition of the ISR (*Sidrauski et al., 2015*) with proteosomal inhibitors such as bortezomib, such a regimen may find an Achilles heel for this family of difficult-to-treat cancers.

### Ideas and speculation

Our findings are also likely to have relevance for other diseases, including neurodegenerative diseases, which are driven by or associated with protein aggregates (*Soto and Pritzkow, 2018*). To date, the function of the ISR in neurodegenerative diseases has been attributed largely to the emergence of alternative translation pathways and to ATF4 transcriptional activity that drives expression of cytoplasmic chaperones and balances cell survival and cell death (*Bond et al., 2020*; *Costa-Mattioli and Walter, 2020*). In SCC cells, although it is possible that chaperones may be upregulated indirectly through ISR-driven transcription factors, these proteins did not emerge as major translational targets of the ISR in our ribosome profiling analysis. Moreover, since cells have parallel pathways, most notably the heat shock response, that can upregulate chaperones when needed (*San Gil et al., 2017*), it seems unlikely that chaperone induction would be ISR's sole function in maintaining proteostasis.

The necessity of an ISR-driven pathway for preserving centrosome dynamics during proteotoxic stress raises an important question regarding centrosome function in neurodegenerative disorders. If ISR-driven translation of centrosomal proteins is required to protect microtubule dynamics during stress, it follows that the accumulation of unfolded proteins should negatively impact the MTOC and/or its microtubule-associated dynamics. In Alzheimer's disease and other tauopathies, the microtubule-associated protein Tau is the main driver of the aggregates of neurofibrillary tangles that ensue, and this alone is likely to negatively impact microtubule-mediated trafficking in neurons (*Ballatore et al., 2007*). In addition, misfolded protein aggregates often inadvertently sequester properly folded cellular proteins, such as p62 and the disaggregase p97/VCP, which can exacerbate their toxic effects on cells (*Donaldson et al., 2003*; *Olzscha et al., 2011*; *Yang and Hu, 2016*). It seems plausible that misfolded protein aggregates might either sequester low-complexity centrosomal or pericentrosomal proteins (*Woodruff et al., 2017*), or structurally interfere with MTOC function. In fact, one study found that proteotoxic stress does indeed inhibit centrosome function in neurons (*Didier et al., 2008*), raising the question as to whether the ISR might also be involved in maintaining the MTOC of long-lived, nondividing neurons, which face a need for uniquely long-distance microtubule-mediated trafficking.

In summary, our work supports a model in which cancer cells rely upon the ISR to restore proteostasis following protein aggregate stress. By redirecting the translational machinery towards synthesizing proteins involved in enlarging the MTOC and bolstering microtubule dynamics, the ISR aids in the intracellular microtubule-mediated trafficking necessary to assemble aggresomes at the MTOC, where they can be efficiently targeted to the perinuclear protein degradation machinery.

## Materials and methods

### Key resources table

| Reagent type (species) or resource | Designation | Source or reference | Identifiers | Additional information |
|---|---|---|---|---|
| Cell line (*Mus musculus*) | S51 | This paper | - | ISR-competent (HRasG12V TgfbrII D/D) |

*Continued on next page*

*Continued*

| Reagent type (species) or resource | Designation | Source or reference | Identifiers | Additional information |
|---|---|---|---|---|
| Cell line (*M. musculus*) | S51A | This paper | - | ISR-null (HRasG12V TgfbrII D/D) |
| Cell line (*M. musculus*) | TgfbrII fl/fl | *Yang et al., 2015* | - | Keratinocytes |
| Cell line (*M. musculus*) | HRasG12V TgfbrII fl/fl | *Yang et al., 2015* | - | Keratinocytes |
| Cell line (*M. musculus*) | HRasG12V TgfbrII D/D | *Yang et al., 2015* | - | Keratinocytes |
| Recombinant DNA reagent | pLKO-PGK-eIF2a(S51)-Myctag-P2A-NeoR | This paper | - | LV construct to overexpress WT eIF2a |
| Recombinant DNA reagent | pLKO-PGK-eIF2a(S51A)-Myctag-P2A-NeoR | This paper | - | LV construct to overexpress ISR-null (S51A) eIF2a |
| Recombinant DNA reagent | pLKO-TRE-ATF5-GFPtag_UBC-rtTA3-IRES-PuroR | This paper | - | LV construct to overexpress ATF5 |
| Recombinant DNA reagent | pLKO-TRE-RFP-shATF5.1 UBC-rtTA3-IRES-PuroR | This paper; original *Fellmann et al., 2013* | - | LV construct to inducibly knockdown ATF5 |
| Recombinant DNA reagent | pLKO-TRE-RFP-shATF5.2 UBC-rtTA3-IRES-PuroR | This paper; original *Fellmann et al., 2013* | - | LV construct to inducibly knockdown ATF5 |
| Antibody | P-eIF2α (rabbit polyclonal) | Invitrogen | 44-728G | 1:1000 |
| Antibody | Ubiquitin (rabbit polyclonal) | CST | 58395 | 1:1000 |
| Antibody | ATF5 (mouse monoclonal) | Santa Cruz | Sc377168 | 1:1000 |
| Antibody | ATF4 (rabbit monoclonal) | CST | 11815 | 1:1000 |
| Antibody | p62 (rabbit monoclonal) | CST | 7695 | 1:250 |
| Antibody | PCNT (rabbit polyclonal) | Abcam | Ab4448 | 1:250 |
| Antibody | ATF5 (rabbit monoclonal) | Abcam | Ab184923 | 1:200 |
| Antibody | Cl Casp 3 (rabbit polyclonal) | CST | 9661 | 1:200 |
| Antibody | α-Tubulin (rabbit monoclonal) | Bio-Rad | MCA77G | 1:500 |
| Antibody | β-Actin (mouse monoclonal) | CST | 3700 | 1:1000 |
| Antibody | Puromycin (mouse monoclonal) | MilliporeSigma | MABE343 | 1:1000 |
| Commercial assay or kit | RiboCop v2 | Lexogen | 144 | |
| Commercial assay or kit | SiR-tubulin | Spirochrome | SC002 | |
| Sequence-based reagent | eIF2a gRNA | Target sequence ATATTCCAACAAGCTGACAT | IDT | |
| Peptide, recombinant protein | Cas9 | IDT | 1081058 | |
| Chemical compound, drug | Sodium arsenite | VWR | BJ35000 | |
| Chemical compound, drug | Tunicamycin | Sigma-Aldrich | T7765 | |
| Chemical compound, drug | Bortezomib | Calbiochem | T7765 | 5043140001 |
| Chemical compound, drug | Cyclohexamide | MilliporeSigma | T7765 | C7698 |
| Chemical compound, drug | Rhodamine phalloidin | MilliporeSigma | R415 | |

*Continued on next page*

*Continued*

| Reagent type (species) or resource | Designation | Source or reference | Identifiers | Additional information |
|---|---|---|---|---|
| Commercial assay or kit | Annexin-V PE | Thermo Fisher | A35111 | |
| Commercial assay or kit | Annexin-V AF647 | Thermo Fisher | A23204 | |
| Chemical compound, drug | Salubrinal | Sigma | SML0951 | |
| Chemical compound, drug | PERK Inhibitor I, GSK2606414 | Sigma | 516535 | |
| Chemical compound, drug | Bafilomycin A1 from *Streptomyces griseus* | MilliporeSigma | B1793 | |
| Chemical compound, drug | Nocodazole | Sigma | M1404 | |
| Chemical compound, drug | Paclitaxel | MilliporeSigma | T7191 | |

## CRISPR cloning

Lentiviral particles containing the gene replacement construct, pLKO-PGK-eIF2α-mycTag-P2A-NeoR, were prepared by transfecting the lentiviral plasmid along with packaging plasmids into HEK293T cells, and viral supernatant was collected 48 hr post transfection. This construct was integrated into the genome of a clonal, parental primary SCC mouse line by incubating 100 µL with 0.1 mg/mL polybrene for 8 hr. 48 hr later, cells with the integrated construct were selected with 0.5 mg/mL neomycin selection for 2 days. Following selection, the endogenous allele was targeted for deletion using CRISPR-Cas9 RNP particles. The replacement allele had a synonymous mutation in the PAM site, rendering it resistant to this CRIPSR construct. CRISPR-Cas9 RNP particles targeting the endogenous allele were prepared as follows: eIF2α gRNA (target sequence: ATATTCCAACAAGCTGACAT) was designed using GuideScan software (*Perez et al., 2017*) and complexed with ATTO550-tracrRNA and Cas9. All reagents were acquired from IDTdna's 'AltR system.' Duplexed gRNA:tracrRNA was prepared by mixing 1 µM of each component in IDTdna duplex buffer, heated to 95°C in a thermocycler and annealed by gradually lowering the temperature to 25°C at a rate of 0.1°C/s. Duplexed gRNA:tracrRNA was complexed with Cas9 by mixing 1 µM of the duplex with 1 µM Cas9 in OptiMEM (Thermo Fisher) and incubating at room temperature for 5 min.

RNP complexes then were transfected into 60% confluent 12-well plates using RNAiMAX as follows: 30 µL of 1 µM RNP complexes were mixed with 4.8 µL RNAiMAX in 335 µL of optiMEM and RNP-lipid complexes were allowed to form for 15 min at room temperature. At the end of the incubation, 400 µL of complexes were added dropwise to cells, and media were changed 16 hr later. 48 hr after transfection, single cells were isolated using FACS as follows: cells were dissociated with trypsin (Gibco) and resuspended in 500 µL of FACS buffer (PBS supplemented with 5% FBS and 5 µM EDTA), and single, ATTO-550-positive cells were sorted using BD FACSAria cell sorter into wells of 96-well plates containing 100 µL of 50-50 mixture of fresh media and conditioned media. Clones that grew to confluency were transferred to 12-well plates, and following growth to confluency in 12-well plates, cells were dissociated with trypsin and frozen in freezing media supplemented with 10% FBS and 10% DMSO. At this stage, a small aliquot of cells (75% of the plate) was lysed in 200 µL of QuickExtract DNA Extraction solution (Lucigen) and gDNA was prepared by heating to 65°C for 10 minu followed by heat inactivation at 95° C for 2 min. These gDNA samples were used for further analysis.

## NGS analysis of CRISPR outcomes

Knockout of the endogenous allele was evaluated using primers targeting a 300 basepair region of genomic DNA with the targeted locus in the middle of the amplicon. Primers had 5' overhangs with sequences compatible with the Illumina Nextera XT index primers (R: overhang: GTCTCGTGGGCTCGGAGATGTGTATAAGAGACAG; L overhang: TCGTCGGCAGCGTCAGATGTGTATAAGAGACAG). Amplicons were generated with 1 µL of input gDNA and the NEB Phusion kit according to the manufacturer's instructions, and amplicons were isolated using Agencout Ampure XP beads (Beckman Coulter). A second barcoding PCR was performed using Nextera XT index primers as follows: 2 µL of cleaned amplicons were used as input, primers were added so that each isolated clone had a unique combination of left and right barcodes, and barcodes were added using an eight-cycle PCR reaction with 55°C annealing temp, again with the NEB Phusion kit, and the barcoded amplicons were cleaned and primer dimers were removed using Ampure XP beads. Amplicons were normalized to the same

concentration, pooled, and sequenced using a single Illumina MiSeq Nano lane using the 250 base-pair, paired end kit. Demultiplexed reads were analyzed and screened for indels using the RGEN Cas Analyzer (http://www.rgenome.net). A KO clone was confirmed if the only reads detected in that sample were indels that would create a frameshift.

## Cell culture

Primary murine SCC cells were generated and cultured in E medium supplemented with 15% FBS and 50 mM CaCl$_2$ as previously described (*Yang et al., 2015*). Cells were passaged three times per week, and passage numbers were maintained counting from the point of cell line generation. Frozen cell stocks were generated by freezing cells in complete media supplemented with 10% additional FBS and 10% DMSO. Primary keratinocytes used in this study have been previously characterized (*Yang et al., 2015*). Cells routinely tested negative for mycoplasma contamination. ATF5-GFP-overexpressing cells were generated by infecting ISR-null cells with ATF5-GFP lentivirus in the presence of 5 µg/mL polybrene. Infected cells were selected using 2 µg/mL puromycin for 5 days. Induction of ATF5 was induced using 1 µg/mL doxycycline at the time of plating for each experiment. ISR-competent cells with ATF5 knockdown were generated following the same protocol. Knockdown vectors were generated using the miR-E system (*Fellmann et al., 2013*).

## Proliferation and cell viability assays

For proliferation assays, 2500–5000 cells were plated per well in clear-bottom, black optical 96-well plates (Nunc) and allowed to attach overnight. A baseline plate was collected the next day as a zero hour sample, and then plates were collected at 24, 48, and 72 hr after this timepoint. At the time of collection, media were washed and cells were fixed in 4% PFA in PBS for 10 min at room temperature. After fixation, PFA was washed with PBS and cells were stored in PBS at 4°C until the end of the experiment. Following collection of the final plate, nuclei were stained in all samples using 1 µg/mL DAPI in PBS for 5 min at room temperature. DAPI was washed and replaced with PBS, and nuclei were imaged on a Biotek Cytation 5 high-content imager, and cells were counted using Gen5 software. To measure cell death, single-cell-suspensions were washed once with PBS, stained with 3 µL of AnnexinV-PE or Annexin-V AF647 conjugates (Thermo Fisher) and 0.1 µg/ml DAPI (Thermo Fisher) in 100 µL of 1× Annexin Binding Buffer (Thermo Fisher) for 15 min at room temperature. Data were collected using a BD LSR Fortessa X20 flow cytometer and analyzed using FlowJo software.

## Measurement of translation rates

Cells were plated to 75% confluency in six-well plates. The next day cells were treated with 50 µM sodium arsenite, 100 ng/mL tunicamycin or 100 nM bortezomib, or a vehicle control and incubated at 37°C for 6 hr. At this time, 20 µM puromycin was supplemented to the media, and cells were incubated for 30 min. Cells were then lysed in RIPA buffer, and translation rates were evaluated by immunoblotting for puromycilated peptides using an anti-puromycin antibody.

## Microtubule nucleation assay

Cells were plated to 50% confluency in glass slides (Millicell EZ, MilliporeSigma) and treated with 100 nM bortezomib or a vehicle control for 6 hr followed by a PBS wash and replacement with fresh media. 4 hr later, the media were supplemented with 13 µM nocodazole, and cells were incubated at 37°C for 20 min. At the end of the incubation, slides were washed and media were replaced with fresh, nocodazole-free media, and microtubules were allowed to recover during a 2 min incubation at room temperature. At this time, cells were fixed in 4% PFA for 10 min at room temperature. Additional control slides without nocodazole or with nocodazole and no recovery were fixed as controls. Slides were processed for immunofluorescence targeting α-tubulin, pericentrin, and $\lambda$-tubulin, immunofluorescence signal was imaged by confocal microscopy, and microtubule nucleation rates were evaluated by quantifying the relative α-tubulin signal intensity within the centrosome region, which was defined using 3D volumetric assessment (Imaris) of pericentrin-positive volumes.

## Live imaging of microtubule dynamics

Cells were plated in plastic-bottom optical slides (Ibidi 80826). After 24 hr, cells were treated with 100 nM bortezomib for 6 hr, followed by a PBS wash and a 4 hr recovery period in ELow media.

1 hr prior to imaging, fresh media containing 100 nM SiR-tubulin (Spirochrome SC002) and 10 µM verapamil (an efflux pump inhibitor) were added. Cells were placed in a spinning-disk Dragonfly chamber with prewarmed temperature set at 37°C and 5% $CO_2$. Prior to imaging, cells were incubated with 13 µM nocodazole for 20 min, washed once with PBS and fresh media with 100 nM SiR-tubulin, and 10 µM verapamil was added. Images were acquired with a 1 µm z-stack every 2 s for 7 min with a 63× oil objective, using Fusion imaging software. Videos were analyzed using Imaris software. Images from frames 1, 11, 21, 31, 41, 51, and 61 were extracted, and total volume of SiR-tubulin signal coming from centrosomes (as identified at frame 1) was calculated per each cell. For videos acquired on different days, frames were aligned on pseudotime, based on the start time of microtubule growth in S51 SCC cells. Experiment was performed in two independent biological replicates (each with two technical replicates).

## Scratch assay

Cells were plated in plastic-bottom optical slides (Ibidi 80826) and allowed to reach confluency in 48 hr. At this time, cells were treated with 100 nM bortezomib or a vehicle control for 6 hr followed by PBS wash and replacement with fresh, drug-free media. At the time of wash, scratch wounds were manually created by gently scaping cells using a rubber cell scraper. Movement of wound-edge SCC cells was evaluated using time-lapse confocal microscopy of GFP signal (marking all cells) with images acquired every 2 min for 8 hr. Scratch closure was evaluated using Imaris software to measure the percentage of scratch closed by leading edge cells over the course of the experiment.

## Tumor allografting

SCC allografts were generated by intradermally injecting $1 \times 10^5$ SCC cells suspended in a 50:50 mix of PBS and growth-factor-reduced Matrigel (Corning, 356231) in an injection volume of 50 µL. Grafts were generated in the flanks of 6–8-week-old female (*Nude*) mice. Tumor dimensions were measured every 5 days using electronic calipers, and tumor volume was calculated using the formula $V = 0.5 \times \text{length} \times \text{width}^2$. For tumor growth experiments, sample sizes of N = 8 were calculated to yield an 80% power to detect a significant (p<0.05) effect size of 50% assuming a standard deviation of 25%, conservative estimates based on past lab experience suggesting targeting ISR-related genes in SCC that yielded larger effect sizes (*Sendoel et al., 2017*). The experiment was performed twice ISR-competent and ISR-null using two different clones for each, with each experiment powered independently to 80%, yielding total sample size of N = 16 when pooling data. Experiments evaluating apoptosis in vivo were set up for 80% power to detect a significant (p<0.05) effect size of 66% with 25% SD, which yielded a minimum sample size of N = 4.

## FACS isolation of ex vivo tumor cells

To sort SCC cells out of tumors formed from ISR-WT and ISR-null cells lines, day 35 tumors were dissected from the skin and finely minced in 0.25% of collagenase (Sigma) in HBSS (Gibco) solution. The tissue pieces were incubated at 37°C for 20 min by gently shaking. After a single wash with ice-cold PBS, the samples were further digested into single-cell suspension in 0.25% trypsin/EDTA (Gibco) for 10 min at 37°C. After neutralization with the FACS buffer (PBS supplemented with 4% FBS, 5 mM EDTA, and 1 mM HEPES), single-cell suspensions was then centrifuged, resuspended, and strained before preparing for staining. A cocktail of Abs for surface markers at the predetermined concentrations (CD31-APC 1:100, BioLegend; CD45-APC 1:200, BioLegend, CD117-APC 1:100, BioLegend; CD140a-APC 1:100, Thermo Fisher; CD29-APCe780, 1:250, Thermo Fisher, BioLegend) was prepared in the FACS buffer with 100 ng/mL DAPI. Sorting was performed using a BD FACSAria equipped with FACSDiva software to isolate a population of cells that was GFP-positive (pan-SCC), DAPI-negative (live), and APC-negative (dump gate to exclude immune cells, endothelial cells, and fibroblasts), and new cell lines were established from bulk populations of sorted cells.

## Immunofluorescence/histology

For immunofluorescence of tumors, samples were fixed in 4% PFA for 1 hr at room temperature, dehydrated in 30% sucrose overnight at 4°C, and mounted into OCT blocks and frozen. 14-µm-thick sections were cut using a Leica cryostat deposited onto SuperFrost Plus slides (VWR). For immunofluorescence of cells in culture, cells were plated on glass slides (Millicell EZ, MilliporeSigma) coated with

human plasma fibronectin (MilliporeSigma) diluted to 100 µg/mL in PBS. At the conclusion of experiments, samples were fixed with 4% PFA for 10 min at room temperature. Samples were permeabilized with 0.3% Triton-X100 in PBS and blocked using 2.5% normal donkey serum, 2.5% normal goat serum, 1% BSA, 2% fish gelatin, and 0.3% Triton X-100 in PBS. Primary antibodies were applied in blocking buffer overnight at 4°C. Samples were washed with 0.1% Triton X-100, and secondary antibodies with Alexa 488, Alexa 594, and Alexa 647 were applied for 1 hr at room temperature in blocking buffer containing 1 µg/mL DAPI. Slides were washed with 0.1% triton and mounted using Prolong Diamond Antifade Mountant with DAPI (Thermo Fisher).

## Microscopy and image analysis

Microscopy of tumors and ×40 images of aggresomes and spreading cells were performed using an Axio Observer Z1 epifluorescence microscope equipped with a Hamamatsu ORCA-ER camera (Hamamatsu Photonics) and with an ApoTome.2 (Carl Zeiss) slider using a ×20 air, ×40 oil, or ×63 oil objective. The ×63 confocal microscopy images were collected on an Andor Dragonfly spinning disk imaging system with a Leica DMi8 Stand and cMOS Zyla camera. Images were analyzed in Fiji or Imaris. For fluorescence intensity measurements of tumors, GFP masks were generated and signal was measured within the mask. Aggresomes were manually counted as discrete p62-positive juxtanuclear puncta on maximum intensity Z-projections. Cell dimensions were calculated using the length measuring tool, and cell spreading was evaluated manually by observing for spread morphology. RGB images were generated with Fiji and saved as TIFF files. For 3D reconstructions and volumetric analyses of MTOCs, Imaris was used to generate 3D images, and volumes were generated to create 3D volumes encompassing the discrete puncta of pericentrin staining. Volume as well as summed pericentrin and γ-tubulin fluorescence intensity were measured within these volumes.

## Electron microscopy

Cells were fixed in a solution containing 4% PFA, 2% glutaraldehyde, and 2 mM $CaCl_2$ in 0.1 M sodium cacadylate buffer (pH 7.2) for 1 hr at room temperature, and then placed at 4°C. Cells were next postfixed in 1% osium tetroxide and processed for Epon embedding; ultrathin sections (60–65 nm) were then counterstained with uranyl acetate and lead citrate, and images were acquired using a Tacnai G2-12 transmission electron microscope equipped with an AMT BioSprint29 digital camera.

## Cell fractionation

Cells were lysed in RIPA buffer (20 mM Tris–HCl, pH 8.0, 150 mM NaCl, 1 mM EDTA, 1 mM EGTA, 1% Triton X-100, 0.5% sodium deoxycholate, 0.1% SDS) containing protease inhibitors (cOmplete Mini, Roche) and phosphatase inhibitors (PhosSTOP), and lysed for 10 min at room temperature. Membrane fraction was pelleted by centrifuging at 1000 × $g$ for 10 min at 4°C. The supernatant (cytosolic fraction) was transferred to new tubes, which were centrifuged at 20,000 × $g$ for 30 min at 4°C. The supernatant (RIPA-soluble fraction) was transferred to a new tube, and the pellets (RIPA-insoluble fractions) were washed with 300 µL of RIPA buffer, centrifuged 20,000 × $g$ for 10 min at 4°C, and then resuspended in 30 µL of 1× LDS-βME by vortexing vigorously and boiling at 98°C for 10 min. The protein concentration of the RIPA-soluble fraction was measured with a BCA (Pierce). RIPA-soluble fractions were mixed into 1× LDS-βME and protein concentration was normalized. The insoluble fractions were normalized by adding 1× LDS-βMe so that the same volume corresponds to the same volume of insoluble-fraction lysate (e.g., insoluble fraction from 100 µg of cell lysate). Samples were run on immunoblots as previously described and probed for ubiquitin signal.

## Western blotting

Cells were lysed with RIPA (20 mM Tris–HCl, pH 8.0, 150 mM NaCl, 1 mM EDTA, 1 mM EGTA, 1% Triton X-100, 0.5% deoxycholate, 0.1% SDS) containing protease inhibitors (cOmplete Mini, Roche) and phosphatase inhibitors (PhosSTOP). Lysates were clarified by centrifugation, and protein concentration of supernatants was evaluated using BCA (Pierce). Protein lysates were normalized, mixed with LDS (Thermo Fisher) and β-mercaptoethanol (Thermo Fisher), and denatured at 95°C for 5 min. Protein was separated by gel electrophoresis using 4–12% or 12% NuPAGE Bis-Tris gradient gels (Life Technologies) and transferred to nitrocellulose membranes (GE Healthcare, 0.45 µm). Membranes were blocked with 5% BSA in TBS supplemented with 0.1% Tween 20, and primary antibodies were stained

overnight in blocking buffer at 4°C or 1 hr at room temperature, and HRP-conjugated secondary antibodies were stained for 1 hr at room temperature. Membranes were washed and then incubated with ECL plus chemiluminescent reagent (Pierce) for 30 s. Chemiluminescent signal was evaluated using CL-XPosure Film (Thermo Fisher).

## Antibodies and counterstains

The following antibodies and dilutions were used for immunoblotting: 1/1000 β-actin (Cell Signaling Technologies, 3700); 1/1000 p-eIF2α (Invitrogen, 44-728G); 1/5000 p-eIF2α (Cell Signaling Technologies, 3597); 1/1000 eif2a (Cell Signaling Technologies, 9722); 1/3000 puromycin (MilliporeSigma MABE343); 1/1000 p62/SQSTM1 (Cell Signaling Technologies, 5114); 1/1000 Atf4 (Cell Signaling Technologies, 11815); 1/1000 ubiquitin (Cell Signaling Technologies, 58395); 1/1000 K48-linked poly-ubiquitin (Cell Signaling Technologies, 8081); 1/1000 Myc-tag (Abcam, ab32); 1/5000 α-tubulin (MilliporeSigma, T5168); 1/1000 P-S6K (Cell Signaling Technologies 2708); 1/1000 total S6K (Cell Signaling Technologies 9202); 1/1000 P-4EBP1-Thr37 (Cell Signaling Technologies 2855); 1/1000 Total 4EBP1 (Cell Signaling Technologies 9644); and 1/1000 Atf5 (Santa Cruz, sc377168). For immunofluorescence, the following antibodies and dilutions were used: 1/500 α6-integrin (BD Biosciences, 555734); 1/500 GFP (Abcam, ab13970); 1/500 E-cadherin (Cell Signaling Technologies, 3195); 1/500 vinculin (MilliporeSigma, V9131); 1/250 p62/SQSTM (Cell Signaling Technologies, 7695); 1/500 α-tubulin (BD Biosciences, MCA77G); 1/500 γ-tubulin (MilliporeSigma, T6557); 1/250 pericentrin (Abcam, ab4448); and 1/200 Atf5 (Abcam ab184923). Nuclei were counterstained with 1 μg/mL DAPI (Thermo Fisher), and F-actin was stained with 1/400 rhodamine or Alexa Fluor-488-conjugated phalloidin (Thermo Fisher).

## Ribosome profiling and analysis

To perform ribosome profiling, we closely followed a recently published protocol (*McGlincy and Ingolia, 2017*). In short, cells were lysed in polysome buffer supplemented with 0.1 mg/mL cyclohexamide. Lysates were treated with 500 U of RNAse I (Epicentre) per 25 μg of RNA (quantified by Qubit Fluorimetry, Thermo Fisher), and RPFs were isolated using sephacryl S400 columns (GE Healthcare) in TE buffer. Ribosomes and RPFs were dissociated in Trizol (Invitrogen) and RNA was collected with protected fragments purified using Zymo Research DirectZol MiniPrep columns. RPFs were purified by running total RNA on a 15% TBE-Urea gel (Thermo Fisher) and cutting the region corresponding to 17–34 nucleotides. RNA was purified and precipitated, dephosphorylated with PNK (NEB), and ligated to preadenylated barcoded linker oligonucleotides, and then unligated linker was digested away with 25 U yeast 5′-deadenylase (NEB) and 5 U RecJ exonuclease (Epicentre). Up to eight libraries were pooled, and rRNA was depleted using the Lexogen Ribocop V2 kit, and reverse transcription of RPF-linker fragments was performed in presence of 20 U Superase-IN (Invitrogen) and 200 U Proto-Script II (NEB). cDNAs were circularized using 100 U CircLigase I ssDNA ligase (Epicentre), and cDNA concentration was quantified by QPCR of cDNA compared to a standard curve of a reference sample. Libraries were amplified with Illumina-compatible barcoded primers and a 10-cycle PCR. DNA of the correct size (~160 bp) was isolated on TBE-PAGE gel, precipitated, and resuspended in TE buffer. Total mRNA samples were prepared in parallel, and mRNA was selected by rRNA depletion, and libraries were prepared using Illumina Ribozero kit.

Ribosome profiling and total RNA libraries were pooled and sequenced on a NovaSeq using the S1, 1 × 100 bp kit. Reads were demultiplexed, trimmed using FastX trimmer, and aligned to the mm10 reference genome using bowtie2. Sequences were counted in bins of 5′UTR, CDS, and 3′UTR as defined using plastid (https://plastid.readthedocs.io/en/latest/). Count data were analyzed using DESeq2 (*Love et al., 2014*), an R package designed for statistical analysis of gene counts generated from Illumina-based sequencing. For expression analysis of ribosome profiling data, only reads in the CDS were included and reads coming from the first 15 codons (45 bp) and last 5 codons (15 bp) were excluded. Additionally, only reads of size between 20–23 bp or 26–32 bp were counted. Gene lists were generated as described in the main text, first by filtering genes with significant differences in RPF read counts, and then genes that changed specifically at the translation level were identified as the subset of filtered genes with TE (TE = normalized RPF/normalized total RNA) fold changes greater than 1.5.

## Data availability

The total mRNA and RPFs datasets generated during the ribosome profiling experiments performed in this study have been deposited to the Gene Expression Omnibus (GEO) repository with accession code GSE193945.

## Acknowledgements

We thank Rockefeller University's Flow Cytometry Resource Center (Svetlana Mazel, director), the Genomics Resource Center (Connie Zhao, director), and the American Association for the Accreditation of Laboratory Animal Care-accredited comparative biology center (R Tolwani, director) for their services. We thank the Epigenomics Core Facility (Yushan Li, director) at Weill Cornell University for sample processing. We also thank members of the Fuchs lab, specifically Lisa Polak for assisting with the engraftments and tumor studies, and Dr. Stephanie Ellis and Dr. Katherine S Stewart for their helpful discussions and for assistance in setting up various experiments performed in this study. BH is a Ruth Kirschstein NIH Predoctoral Fellow (F30CA236239) and a member of the Weill Cornell/Rockefeller/Sloan Kettering Tri-Institutional Medical Scientist Training Program (T32GM007739). NG is an HHMI Jane Coffin Childs Associate. AG is a Damon Runyon Cancer Research Foundation National Mah Jongg League Fellowship (DRG 2409-20). EF is an investigator of the Howard Hughes Medical Institute. The work was also supported by grants from the National Institutes of Health (R01-AR27883 to EF) and the Robertson Foundation.

## Additional information

### Competing interests

Elaine Fuchs: Reviewing editor, eLife. The other authors declare that no competing interests exist.

### Funding

| Funder | Grant reference number | Author |
| --- | --- | --- |
| Howard Hughes Medical Institute | | Elaine Fuchs |
| Ruth Kirschstein NIH Predoctoral Fellow | F30CA236239 | Brian Hurwitz |
| Weill Cornell/Rockefeller/Sloan Kettering Tri-Institutional Medical Scientist Training Program | T32GM007739 | Brian Hurwitz |
| Howard Hughes Medical Institute | Jane Coffin Childs Associate | Nicola Guzzi |
| Damon Runyon Cancer Research Foundation | National Mah Jongg League Fellowship (DRG 2409-20) | Anita Gola |
| National Institutes of Health | R01-AR27883 | Elaine Fuchs |
| Robertson Foundation | | Brian Hurwitz |

The funders had no role in study design, data collection and interpretation, or the decision to submit the work for publication.

### Author contributions

Brian Hurwitz, Conceptualization, Data curation, Formal analysis, Investigation, Methodology, Project administration, Visualization, Writing – original draft, Writing – review and editing; Nicola Guzzi, Data curation, Formal analysis, Investigation, Methodology, Validation, Visualization, Writing – original draft, Writing – review and editing; Anita Gola, Data curation, Formal analysis, Investigation, Methodology, Visualization, Writing – review and editing; Vincent F Fiore, Data curation, Formal analysis,

Investigation, Methodology; Ataman Sendoel, Conceptualization, Investigation; Maria Nikolova, Methodology; Douglas Barrows, Formal analysis; Thomas S Carroll, Formal analysis, Supervision; H Amalia Pasolli, Data curation, Methodology; Elaine Fuchs, Conceptualization, Data curation, Funding acquisition, Supervision, Writing – original draft, Writing – review and editing

**Author ORCIDs**
Brian Hurwitz http://orcid.org/0000-0002-1948-6317
Nicola Guzzi http://orcid.org/0000-0002-3898-8064
Elaine Fuchs http://orcid.org/0000-0002-0978-5137

**Ethics**
Animal experimentation: All animal procedures used in this study are described in our #20066H protocol named Development and Differentiation in the Skin, which had been previously reviewed and approved by the Rockefeller University Institutional Animal Care and Use Committee (IACUC).

**Decision letter and Author response**
Decision letter https://doi.org/10.7554/eLife.77780.sa1
Author response https://doi.org/10.7554/eLife.77780.sa2

## Additional files

**Supplementary files**
• Transparent reporting form

**Data availability**
Sequencing data have been deposited in GEO under accession codes GSE193945.

The following dataset was generated:

| Author(s) | Year | Dataset title | Dataset URL | Database and Identifier |
|---|---|---|---|---|
| Hurwitz B, Guzzi N, Gola A, Fiore V, Sendoel A, Nikolova M, Barrows D, Carrol TS, Pasolli AH, Fuchs E | 2022 | The integrated stress response remodels the microtubule organizing center to clear unfolded proteins following proteotoxic stress | https://www.ncbi.nlm.nih.gov/geo/query/acc.cgi?acc=GSE193945 | NCBI Gene Expression Omnibus, GSE193945 |

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
