## [Editor Report]

This interesting study identifies why or how the integrated stress response pathway regulates cell recovery upon proteotoxic stress, which is especially interesting in cancer cells resistant to proteasome inhibitors. The authors conclude that translation initiation of mRNAs encoding microtubule cytoskeleton, centrosome, and ATF5 proteins is necessary to recover from proteotoxic stress. This article makes a strong contribution to the literature.

---

## [Decision Letter]

**Decision letter after peer review:**

Thank you for submitting your article "The integrated stress response remodels the microtubule organizing center to clear unfolded proteins following proteotoxic stress" for consideration by *eLife*. Your article has been reviewed by 3 peer reviewers, and the evaluation has been overseen by a Reviewing Editor and Marianne Bronner as the Senior Editor. The reviewers have opted to remain anonymous.

The reviewers have discussed their reviews with one another, and the Reviewing Editor has drafted this to help you prepare a revised submission. The reviews are overall positive about the work's suitability for *eLife*, but there is a need to address the reviewers' criticisms that will require extensive revisions. Due to the relevance of all the points in strengthening the work, all the reviewers' comments should be addressed. We look forward to receiving your revised manuscript,

*Reviewer #1 (Recommendations for the authors):*

1) What happens to PERK activity in the context of S51A+bortezomib? It is very likely that PERK is activated which raises the question if the phenotypes the authors observe are mediated by an ISR-independent, but PERK-dependent mechanism. This could be answered by generating PERK KO cells in the S51A background.

2) Lines 301-303 – What are the mRNA levels of the 199 translationally upregulated mRNAs? Do they change when cells undergo proteotoxic stress? The X-axis of Figure 3C appears to indicate a very wide variation in transcript abundance ranging from log2 2-14. It is prudent for the authors to show this data on a per gene basis to confirm that the changes are specifically at the translation level. In addition, Figures 3E and F – it would be good to show the mRNA levels of each of the 24 genes (a complementary heatmap).

3) Figure 3E – the authors list p-values without a correction for multiple testing (FDR or padj). In my analysis of these 24 genes in DAVID GO, it appears that with the Benjamini test, these BPs and CCs are not statistically significant.

4) Figure 3H and I. The author should establish that the effects they see are truly nucleated at the translational level by showing the mRNA levels of ATF5 between conditions in their RP data (Figure 3H). They should also QPCR for ATF5 for Figure 3I.

5) Lines 493-499 – The RP data presented here does not indicate that the protein levels of these focal adhesion genes are actually down. This should be validated by western or MS.

6) Is ATF5 translation specific to proteotoxic stress? Or it is generally responsive to eIF2alpha phosphorylation? Would be good to determine this using NaAs02 and Tunicamycin. There may be wider implications beyond proteotoxic stress.

7) Why is ATF5 preferentially translated in the context of proteotoxic stress? Is there a functional cis-regulatory element within the 5' UTR that enables context-specific translation?

*Reviewer #2 (Recommendations for the authors):*

The findings from Hurwitz et al. expose a novel mechanism underlying the ISR in cancer cells that will be of interest to people in different fields. The presented data is convincing and statistically robust, clearly demonstrating the existence of the involvement of the MTOC in the process. Although the direct molecular connections between MTOCs and the destruction of the protein aggregates are questions for the future, and there are no significant weaknesses, adding some additional analyses will help to strengthen their conclusions.

1. The authors generated valuable tools to study the ISR response by modifying elF4alpha, a major component of the canonical translational complex. The authors generated elF2alpha-myc and mutant elF2alphaS51-myc (ISR-null) expressing cells. Is the translation process in steady-state conditions comparable to the parental cells? It is indicated in the text (line 146), but the information is missing in Figure 1D.

2. Ribosomal profiling approaches revealed enrichment in centrosomal proteins upon activation of the IRS compared to IRS null cells, suggesting an involvement of the centrosome and microtubule-dependent mechanism in a proficient IRS process. The authors indicate that microtubule dynamics decrease upon activation of the IRS. The authors made this claim after measuring the centrosome size and microtubule length after recovery from depolymerization. The data indeed suggest that microtubule dynamics decrease upon IRS. However, MT dynamics and turnover have not been formally tested. MT turnover analyses and growth at the MT ends would be necessary to validate it fully.

3. The centrosome is the major MTOC, but other MTOCs coexist in the cells. Figure 7C shows the expression of ATF5 in other distinct areas within the cells besides the centrosome. Are the proteins identified in their ribosomal profiling localizing only at the centrosome?

4. Are there any ultrastructural changes in the centrioles upon the induction of the IRS?

*Reviewer #3 (Recommendations for the authors):*

How the cells recovered from proteotoxic stress seems to be dependent on the canonical translation of ATF5, which when overexpressed in mutant S51A cells rescues the in vivo apoptotic and centrosome phenotype to wt cells at comparable levels. Yet the mechanistic link between ATF5, increased protein aggregate formation, increased microtubule and centrosome nucleation or migration is unclear.

In line with the previous question, one major effect of inhibiting the proteasome is blocking mitotic and cell cycle phases transitions due to its major role in the cell cycle. The study finds that using this proteasome inhibitor induces similar effects: differential expression of mRNAs encoding cell cycle, centrosome, or microtubule organizing centre. The question is whether other ISR inducting drugs can induce a similar mechanism or is only proteasome-dependent?

The authors show in the study that proteotoxic stress induced by inhibition of the proteasome using bortezomib activates a distinct translation program necessary for cell recovery which includes microtubule cytoskeleton, centrosome and ATF5 proteins and claim to be a global mechanism to recover from ISR. However, in figure 1D, the authors show that bortezomib induces little ISR induction compared to the mild protein inhibition effects induced upon bortezomib. In fact, tunicamycin and bortezomib or oxidative stress activate different kinases. Tunicamycin and bortezomib activate PERK pathway and oxidative stress activates HRI pathway. Is oxidative stress inducing a similar translation recovery program?

One question that remains unanswered is how cells re-activate canonical translation upon stress that allows them to recover? Is this phosphatase dependent? Can the cells recover from proteotoxic stress by inhibiting this mechanism?

---

## [Author Response]

Essential revisions:Reviewer #1 (Recommendations for the authors):1) What happens to PERK activity in the context of S51A+bortezomib? It is very likely that PERK is activated which raises the question if the phenotypes the authors observe are mediated by an ISR-independent, but PERK-dependent mechanism. This could be answered by generating PERK KO cells in the S51A background.

Following the suggestion of this Reviewer, we monitored PERK activation in S51 and S51A cells. Indeed, S51A show hyperactivation of PERK signaling compared to S51 cells, as shown by increased PERK phosphorylation at Thr980. We included this information in the manuscript in Figure 2—figure supplement 3D. To further investigate the role of PERK signaling in aggregate clearance, we used the PERK inhibitor GSK2606414. By inhibiting PERK signaling in both S51 and S51A cells, we showed that PERK signaling contributes to aggregate clearance, at least in part, independently from the ISR since both cell lines show higher aggregate accumulation after 24 hrs recovery from bortezomib when PERK signaling is inhibited (Figure 2—figure supplement 3D). To further validate these data, we analyzed the number of aggresomes formed by S51 and S51A cells in presence/absence of PERK inhibition. In agreement with our data, inhibiting PERK resulted in a decrease in number of cells with an aggresome in both cell lines as judged by coalescence of p62 signal by immunofluorescence. These data are now presented in Figure 5—figure supplement 1.

That said, when PERK is inhibited, ISR-null cells still retain higher accumulation of ubiquitin positive aggregates in the insoluble fraction when compared to ISR-competent cells. These data rule out the notion that the observed phenotype in ISR-null cells results simply from hyperactivation of PERK signaling. Rather, they imply the existence of both ISR dependent and independent mechanisms underlying the recovery phase. Although our study here focuses on the ISR-dependent mechanisms, we mention that the ISRindependent mechanism may involve PERK-mediated phosphorylation of one of its other targets, NRF2, as previously described (Pajares et al., 2017).

2) Lines 301-303 – What are the mRNA levels of the 199 translationally upregulated mRNAs? Do they change when cells undergo proteotoxic stress? The X-axis of Figure 3C appears to indicate a very wide variation in transcript abundance ranging from log2 2-14. It is prudent for the authors to show this data on a per gene basis to confirm that the changes are specifically at the translation level. In addition, Figures 3E and F – it would be good to show the mRNA levels of each of the 24 genes (a complementary heatmap).

Our strategy for identifying translationally upregulated mRNAs included a filtering step where only mRNAs that showed increased translation efficiency were considered. Since translation efficiency takes into account changes at the transcriptional level, our findings highlight mRNAs whose translation is specifically increased. While we do not exclude that concomitant transcriptional changes might occur, our filtering strategy allows us to focus on those mRNAs where translation is further upregulated over any transcriptional changes.

To further illustrate this, we generated a heatmap (Figure 3—figure supplement 2A) of the 24 ISR translational targets highlighting that increased expression of these mRNAs is not due to their increased transcription upon bortezomib treatment.

3) Figure 3E – the authors list p-values without a correction for multiple testing (FDR or padj). In my analysis of these 24 genes in DAVID GO, it appears that with the Benjamini test, these BPs and CCs are not statistically significant.

As per the Reviewer suggestion, we included Benjamini p-values as calculated by the DAVID GO software. This information is included in Figure 3G.

4) Figure 3H and I. The author should establish that the effects they see are truly nucleated at the translational level by showing the mRNA levels of ATF5 between conditions in their RP data (Figure 3H). They should also QPCR for ATF5 for Figure 3I.

We included new data showing that ATF5 is not transcriptionally induced by bortezomib treatment, including qPCR validation These data are now included in Figure 3—figure supplement 2A and 2C.

5) Lines 493-499 – The RP data presented here does not indicate that the protein levels of these focal adhesion genes are actually down. This should be validated by western or MS.

We agree with this Reviewer that to have definitive proof that focal adhesion components are upregulated western blot analysis should be performed. However, due to high background of the commercially available antibodies for the focal adhesion components identified by our ribosome profiling analysis this was not possible. To avoid over-interpretation of the data, we revised the indicated paragraph (Lines 570-572 of the resubmitted manuscript).

6) Is ATF5 translation specific to proteotoxic stress? Or it is generally responsive to eIF2alpha phosphorylation? Would be good to determine this using NaAs02 and Tunicamycin. There may be wider implications beyond proteotoxic stress.

Following this Reviewer suggestion, we performed western blot analysis of S51 cells after 6 hrs exposure to either sodium arsenite, tunicamycin or bortezomib. Interestingly, ATF5 translational upregulation is specific to proteotoxic stress since neither sodium arsenite nor tunicamycin induced an increase in ATF5 protein expression. These new data are presented in Figure 7A.

7) Why is ATF5 preferentially translated in the context of proteotoxic stress? Is there a functional cis-regulatory element within the 5' UTR that enables context-specific translation?

We thank this Reviewer for raising this question, which we believe is an interesting topic for future research.

Translational control is often encoded in sequences found in the 5’untranslated region (5’UTR) of an mRNA (Hinnebusch et al., 2016). Atf5’s 5’UTR contains at least 2 upstream open reading frames (uORF), which direct its translation. Additionally, alternative splicing is responsible for expression of two isoform of Atf5, which differ in their 5’UTR. Our finding that ATF5 is uniquely induced by proteotoxic stress in SCC stem cells is intriguing, although unraveling the nature of this translational control is out of the scope of the current manuscript. Nevertheless, our results support the notion that the ISR is not merely a on-off pathway, but a finely tuned event that regulates distinct proteins in response to distinct stresses. We speculate that this regulation could be achieved either by differential phosphorylation levels of eIF2a (Figure 1D), by differences in cellular location of the sensing kinase upstream of eIF2a phosphorylation or by activation of different alternative translation factors.

Reviewer #2 (Recommendations for the authors):The findings from Hurwitz et al. expose a novel mechanism underlying the ISR in cancer cells that will be of interest to people in different fields. The presented data is convincing and statistically robust, clearly demonstrating the existence of the involvement of the MTOC in the process. Although the direct molecular connections between MTOCs and the destruction of the protein aggregates are questions for the future, and there are no significant weaknesses, adding some additional analyses will help to strengthen their conclusions.1. The authors generated valuable tools to study the ISR response by modifying elF4alpha, a major component of the canonical translational complex. The authors generated elF2alpha-myc and mutant elF2alphaS51-myc (ISR-null) expressing cells. Is the translation process in steady-state conditions comparable to the parental cells? It is indicated in the text (line 146), but the information is missing in Figure 1D.

Translation rate at steady state is not affected between ISR-competent and ISR-null cells. The information is provided in line 156 of the corrected manuscript and is shown in Figure 1E and Figure 1—figure supplement 1A by comparing puromycin incorporation rate of vehicle-treated S51 and S51A cells. This information has been added also in the figure legend.

2. Ribosomal profiling approaches revealed enrichment in centrosomal proteins upon activation of the IRS compared to IRS null cells, suggesting an involvement of the centrosome and microtubule-dependent mechanism in a proficient IRS process. The authors indicate that microtubule dynamics decrease upon activation of the IRS. The authors made this claim after measuring the centrosome size and microtubule length after recovery from depolymerization. The data indeed suggest that microtubule dynamics decrease upon IRS. However, MT dynamics and turnover have not been formally tested. MT turnover analyses and growth at the MT ends would be necessary to validate it fully.

We thank the Reviewer for this comment. To provide further evidence of decreased microtubule dynamics in ISR-null cells during recovery from proteotoxic stress, we now performed live imaging experiments using a fluorescently labeled microtubule interacting drug, SiR-Tubulin. Measuring microtubule growth rate at steady state proved to be complicated due to the intricate and dense network of microtubule in these cells. However, we successfully measured centrosomal microtubule growth upon recovery from nocodazole treatment. Our new data are presented in Figure 4F, Figure 4-video supplement 1 and Figure 4—figure supplement 4B. From these new results, it is evident that upon recovery from proteotoxic stress ISR-null cells have delayed microtubule growth and significant non-centrosomal contribution to microtubule growth. Notably, these effects are only apparent during recovery from stress, since untreated S51 and S51A cells show similar centrosomal microtubule growth.

3. The centrosome is the major MTOC, but other MTOCs coexist in the cells. Figure 7C shows the expression of ATF5 in other distinct areas within the cells besides the centrosome. Are the proteins identified in their ribosomal profiling localizing only at the centrosome?

Our data strongly support the notion that aggregates converge at the centrosomal MTOC. This is highlighted in figure 5. In fact, by imaging hundreds of cells we consistently observed p62 signal accumulation at a singular cellular location, which co-localizes with pericentrin and the centrosome. These data indicate that protein aggregates are mostly accumulated at the centrosomal MTOC. Interestingly however, results from our live imaging experiments illustrate that, while S51A cells are largely impaired in centrosomal dynamics, non-centrosomal microtubule growth occurs to allow cellular recovery. Taken together these data suggest that while the ISR is needed to protect the centrosomes, ISR-null cells can compensate by upregulating non-centrosomal microtubule dynamics.

4. Are there any ultrastructural changes in the centrioles upon the induction of the IRS?

This is a fascinating question, but not one easily addressed. Hundreds of serial ultrathin sections across cells would need to be taken to address this question, and even then, if the angle of centrioles are not quite within the section plane, this information will be lost. Having done such studies in the past, they are a major undertaking and often the luck of the draw even when the lab has access to an expert in the field. Still, we obtained new electron microscopy images of centrosomes in S51 SCC cells untreated and following 8 hrs of recovery from proteotoxic stress. These new data shows accumulation of aggresomes at the centrosomal locale only after bortezomib treatment (Figure 5—figure supplement 3A-B).

Reviewer #3 (Recommendations for the authors):How the cells recovered from proteotoxic stress seems to be dependent on the canonical translation of ATF5, which when overexpressed in mutant S51A cells rescues the in vivo apoptotic and centrosome phenotype to wt cells at comparable levels. Yet the mechanistic link between ATF5, increased protein aggregate formation, increased microtubule and centrosome nucleation or migration is unclear.

To further validate the mechanistic link between ATF5 and increased aggregate formation we knocked-down ATF5 in ISR-competent cells (Figure 7—figure supplement 1B-C). Strengthening a mechanistic link between ATF5 expression and centrosomal dynamics, loss of ATF5 led to increased protein aggregate formation upon a bortezomib pulse. This was evidenced by increased ubiquitin staining in the insoluble fraction after

24 hrs of recovery from bortezomib (Figure 7—figure supplement 1D) and decreased formation of aggresomes as quantified by number of cells with p62 foci after 6 hrs recovery from bortezomib (Figure 7figure supplement 1E).

In line with the previous question, one major effect of inhibiting the proteasome is blocking mitotic and cell cycle phases transitions due to its major role in the cell cycle. The study finds that using this proteasome inhibitor induces similar effects: differential expression of mRNAs encoding cell cycle, centrosome, or microtubule organizing centre. The question is whether other ISR inducting drugs can induce a similar mechanism or is only proteasome-dependent?

To further dissect the role of the MTOC upon different stresses, we monitored MTOC size and ATF5 levels upon different stresses (i.e. oxidative stress and ER stress). Remarkably, ATF5 induction and MTOC enlargement were specific to proteotoxic stress response, indicating that the ISR can distinctly fine-tune gene expression in response to different stresses (Figure 4C, Figure 4—figure supplement 2A and Figure 7A). These data suggest that protecting centrosomal dynamics is particularly important in response to proteotoxic stress as a mechanism to maintain microtubule dynamics while concentrating aggregates at a unique cellular localization.

The authors show in the study that proteotoxic stress induced by inhibition of the proteasome using bortezomib activates a distinct translation program necessary for cell recovery which includes microtubule cytoskeleton, centrosome and ATF5 proteins and claim to be a global mechanism to recover from ISR. However, in figure 1D, the authors show that bortezomib induces little ISR induction compared to the mild protein inhibition effects induced upon bortezomib. In fact, tunicamycin and bortezomib or oxidative stress activate different kinases. Tunicamycin and bortezomib activate PERK pathway and oxidative stress activates HRI pathway. Is oxidative stress inducing a similar translation recovery program?

To address this concern, we quantified the levels of eIF2a phosphorylation upon different stresses. In line with the level of protein synthesis rate inhibition (Figure 1E and Figure 1—figure supplement 1A), bortezomib induces mild but significant activation of the ISR. To further evaluate whether sodium arsenite and tunicamycin would activate a similar translation program, we assessed ATF5 levels and MTOC size upon different stresses. Our new data suggest that the identified translation program is unique to bortezomib, since both sodium arsenite and tunicamycin failed to upregulate ATF5 and to affect MTOC size (Figure 4C, Figure 4—figure supplement 2A and Figure 7A).

One question that remains unanswered is how cells re-activate canonical translation upon stress that allows them to recover? Is this phosphatase dependent? Can the cells recover from proteotoxic stress by inhibiting this mechanism?

To address the requirement of eIF2a dephosphorylation for recovery from proteotoxic stress, we inhibited eIF2a phosphatases by using the well-established drug salubrinal. In agreement with a requirement for eIF2a dephosphorylation during the recovery from stress, salubrinal-treated S51 cells failed to clear protein aggregates, as shown by accumulation of Ubiquitin staining in the insoluble fraction after 24 hrs of recovery from proteotoxic stress (Figure 2—figure supplement 3B). Additionally, salubrinal treatment also prevented the increase in MTOC size (Figure 4—figure supplement 2B), suggesting that the response to proteotoxic stress is dynamically regulated and requires both an initial translational upregulation of microtubule components via the ISR and a subsequent recovery of cellular function mediated via inhibition of the ISR.